



# Patterns of wintertime Arctic sea ice leads and their relation to winds and ocean currents

Sascha Willmes[1,2], Günther Heinemann[2], and Frank Schnaase[3]

[1]Earth Observation and Climate Processes, Trier University
[2]Dpt. Environmental Meteorology, Trier, University
[3]Alfred Wegener Institute for Polar and Marine Research, Bremerhaven, Germany

**Correspondence:** Sascha Willmes (willmes@uni-trier.de)

**Abstract.** We use a novel sea-ice lead climatology based on satellite observations with $1\,\text{km}^2$ spatial resolution to identify predominant patterns in Arctic wintertime sea-ice leads. The causes for the observed spatial and temporal variabilities are investigated using ocean surface current velocities and eddy kinetic energies from an ocean model (FESOM) and winds from a regional climate model (CCLM) and ERA5 reanalysis, respectively. The presented investigation provides clear evidence for

the influence of ocean depth and associated currents on the mechanic weakening of sea ice and the accompanied occurrence of sea-ice leads with their characteristic spatial patterns. While the ocean influence on lead dynamics acts on a rather long-term scale (seasonal to inter-annual), the influence of wind appears to trigger sea-ice lead dynamics on shorter time scales of weeks to months and is largely controlled by individual events causing increased divergence.

## 1 Introduction

The fact that sea ice is mobile and exposed to ocean currents and winds causes it to be under constant tractions from above and below, which induces not only sea ice motion, but also internal stress and mechanical weakening. Therefore, sea ice exhibits a distinct dynamical character, which is mostly expressed through the formation of sea-ice leads in the divergent domain (e.g., Feltham, 2008). Knowledge about the variability and spatial distribution of leads is crucial as they promote a very strong exchange of heat and moisture between the relatively warm ocean and the cold winter atmosphere (Marcq and Weiss, 2012;

Lüpkes et al., 2008; Heinemann et al., 2022). Considering the observed changes in Arctic sea-ice extent and the projected trends, understanding the dynamics of leads is key to get a better insight into the feedbacks of the Arctic climate system (e.g. Wang et al., 2016; Zhang et al., 2012; Rheinlaender et al., 2022). Large-scale and high-resolution sea-ice deformation data are also important for improving short-term and seasonal sea-ice forecasts (Korosov et al., 2022; Nguyen et al., 2009). The potential of sea ice leads for preconditioning summer Arctic sea ice has been discussed (Zhang et al., 2018) and leads have

been recognized as a source of global methane, mercury and ozone emissions (Kort et al., 2012; Moore et al., 2014). The recurrence of leads and their spatial distribution are valuable diagnostic parameters for the sea-ice drift (e.g. Spreen et al., 2017; Kwok et al., 2013) and represent an essential habitat for marine mammals and birds (Stirling, 1997). Knowledge about the variability of sea-ice lead dynamics provides crucial information about the exchange of heat, the potential formation of new ice as well as the release of particles into the atmosphere (Creamean et al., 2022; Hartmann et al., 2022). Therefore,





an identification and explanation of the factors that control sea-ice weakening and the associated lead patterns is key for a comprehensive understanding of air - sea ice – ocean interactions. Hence, especially in light of the observed trends in Arctic sea-ice extent (Stroeve and Notz, 2018), and with respect to the projected development of the Arctic climate system (Notz et al., 2020), the structure and dynamics of leads represent essential information for global change monitoring. While patterns in the spatial distribution of leads have recently been identified for both hemispheres with distinct spatial patterns (Reiser et

al., 2019; Willmes and Heinemann, 2016) the driving mechanisms for the profound variability in wintertime sea-ice dynamics are yet to be explained (e.g. Liu et al., 2022, Arthun et al., 2019; Hegyi and Taylor, 2017). The effect of warm air intrusions, extratropical atmospheric circulation and downward infrared radiation on the overall Arctic warming and sea-ice decline have been discussed (Warner et al., 2020; Park et al., 2015; Woods and Caballero, 2016) and first insights into the role of ocean currents on predominant lead occurrences, i.e. the Antarctic Slope Current in the Southern Ocean and the Arctic Boundary

Current, were given by Reiser et al., 2019 and Willmes and Heinemann, 2016, respectively.

In this paper we will first give an overview of the used data (Ch. 1) and then identify regions where leads are forming most frequently using a novel sea-ice lead climatology (Ch. 2). In Chapter 3, we show how predominant lead patterns in wintertime (Nov-Apr) Arctic sea ice as well as trends and anomalies have developed over the winters from 2002/2003 to 2020/2021 for different regions in the Arctic. Subsequently, we identify the influence of ocean floor topography and associated ocean currents

on spatial lead patterns by regional examples. Lastly, Chapter 3 shows how winds, especially the wind field divergence, has triggered temporal lead dynamics in some regions in the Arctic. The results are discussed in Chapter 4 before a comprehensive summary is given in Chapter 5.

## 2 Data and Methods

### 2.1 Sea-ice lead data

We use pan-Arctic daily binary lead maps from Moderate Resolution Imaging Spectroradiometer (MODIS) satellite infrared imagery at a resolution of 1 km$^2$ (data source: Reiser et al., 2020) to derive spatial and temporal lead patterns and a lead climatology for the Arctic Ocean. The daily grids contain one out of 4 basic categories per pixel and day including sea ice, clouds, artefacts and leads. The lead category indicates that the respective grid point was found to exhibit a significant positive surface temperature anomaly with respect to the surrounding sea-ice area in a kernel of 50x50 km$^2$. Artefacts can be considered

as an extension to the MODIS cloud mask as they represent a potential lead detection with, however, large retrieval uncertainty. The metrics and filtering mechanisms that apply to separate sea ice from leads and leads from artefacts, respectively, are described in detail in Reiser et al. (2020).

### 2.2 Bathymetry

To compare our data with ocean depth we use Version 4.0 of the International Bathymetric Chart of the Arctic Ocean (IBCAO)

Grid (Jakobsson et al., 2020). The data were acquired from gebco.net in Polar Stereographic projection co-ordinates (meters),



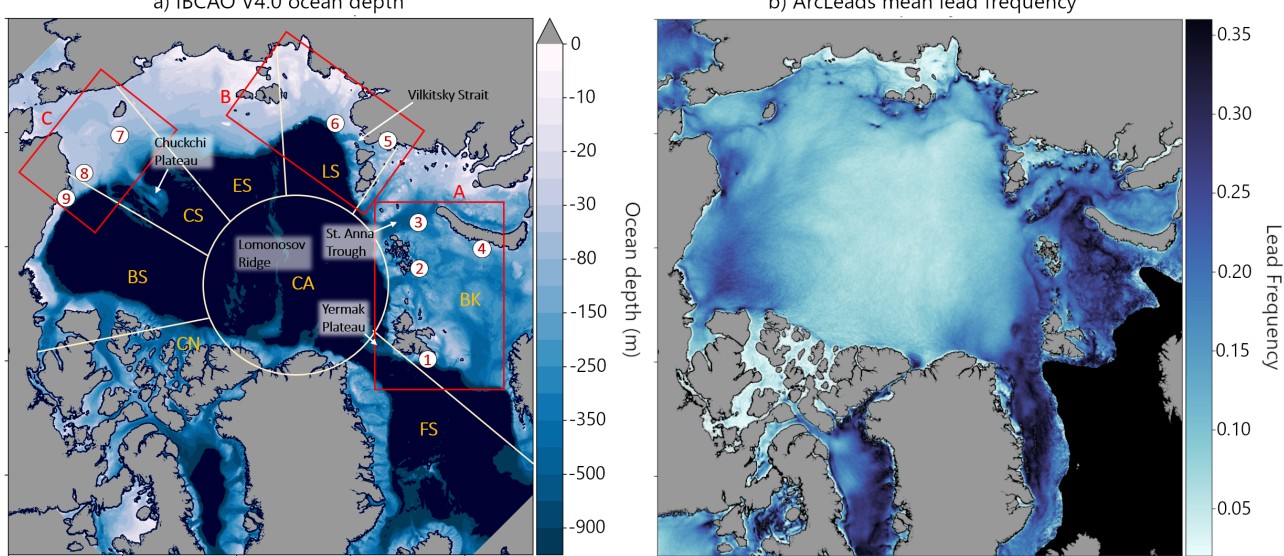

**Figure 1.** a) IBCAO V4.0 ocean depth (Jakobsson et al., 2020) and overview of the Arctic Ocean, analysed sub-regions and points of interest. Red boxes indicate regions used in Figures 3, 4 and 5., b) Pan-Arctic mean sea-ice lead frequencies for the months of NOV-APR in the period 2002/03 – 2020/21 (Reiser et al., 2020)

true scale set at 75°N. We re-projected the data to WGS 84 / NSIDC Sea Ice Polar Stereographic North (EPSG code: 3413) to match with our lead climatology.

## 2.3 Atmospheric data

Atmospheric data are taken from simulations of the non-hydrostatic regional climate model COnsortium for Small-scale MOdel – Climate Limited area Mode (COSMO-CLM or CCLM, Steger and Bucchignani, 2020). CCLM was adapted to polar regions by implementing a two-layer sea ice model and new parameterizations for the atmospheric boundary layer (Heinemann et al. 2021). CCLM has been used for several studies of air/sea-ice/ocean interactions and boundary layer processes in polar regions (e.g., Gutjahr et al. 2016; Kohnemann and Heinemann 2021, Heinemann et al. 2022). CCLM is used with a horizontal resolution of 15 km for the whole Arctic (C15). Initial and boundary data are taken from ERA-Interim reanalyses (Dee et al., 2011). The model is used in a forecast mode with daily reinitialization. Model output is available every 1 h. Sea ice concentration is taken as daily data from AMSR2 data (Spreen et al., 2008). Sea ice thickness is prescribed daily from interpolated Pan-Arctic Ice Ocean Modeling and Assimilation System (PIOMAS) fields (Zhang und Rothrock, 2003). For the present study, monthly averages of wind field data at 10m are used. C15 data are available for the whole Arctic for 2002-2016. As a second atmospheric data set, we use ECMWF ERA5 data (Hersbach et al., 2020), which have a horizontal resolution of about 30 km. We here use monthly averages of wind speed, horizontal wind components and wind divergence for the winter months November to April in the period 2002 to 2021 downloaded from the Copernicus Climate Data Store (https://cds.climate.copernicus.eu/).





## 2.4 Ocean data

Ocean data of the surface current velocity (F_SCV) and Eddy Kinetic Energy (F_EKE) are from simulations of the Finite Elements Sea ice – ice-shelf – Ocean Model (FESOM) version 1.4 (Wang et al., 2014). FESOM was used in the same version as in Sidorenko et al. (2019), but with a different model grid. FESOM runs on a specialized grid with a global horizontal resolution of approximately 1°, which is refined to 4.5 km in the Arctic and 2 km for the Laptev Sea. Vertically it is structured into 47 layers with a resolution of 10 m in the upper 100 m and increased layer density near the bottom. Depth data is taken from the 1 minute version of the RTopo-2 dataset (Schaffer et al. 2016). The model is initialized with temperature and salinity data from the World Ocean Atlas (Levitus et al. 2013), river runoff is from the Japanese 55-year Reanalysis or JRA55 (Suzuki et al., 2018). On the global scale, FESOM is driven by ERA-Interim reanalyses (Dee et al. 2011) for the period 1987-2016. The model output is available as monthly means, with selected variables being available as daily means. We here use FESOM data for the Arctic for the period 2002-2016.

## 2.5 Data processing

Based on the method described in Reiser et al. (2020) and the resulting daily binary lead maps we calculate monthly lead frequencies as well the absolute mean lead frequency for the months of November to April for the winters of 2002/03 to 2015/16 (short period) and 2002/03 to 2020/21 (long period), respectively. The short period is used to allow for comparison with FESOM and C15 simulation results, which are available only up to April 2016. The long period is computed to make use of the full lead data set and allow comparisons with IBCAO ocean depth and ERA5 wind divergence. Two quantities are calculated from the daily binary lead maps. The lead frequency (LFQ) is a temporally integrated quantity that indicates on how many days during a specified period a pixel is found to be covered by a lead, relative to number of the available clear-sky observations (Willmes and Heinemann, 2016). Thereby we obtain e.g., monthly, annual and total lead frequencies for the Arctic. Lead fraction area (LFA), in contrast, is a spatially integrated lead quantity that represents the fraction of a specified area that was covered by leads, i.e., the number of lead pixels over the sum of lead and sea ice pixels. If more than 50% of the given area are covered by clouds or artefacts (Reiser et al., 2020), the lead fraction is flagged "cloud covered". The wind divergence from C15 data was calculated from horizontal wind components at 10m. We also derived deformation and shear (Spreen et al., 2017) from the C15 wind data. Results for deformation and shear are, however, not shown here because no correlation was found between lead dynamics and these two parameters. For ERA5, we directly use the provided monthly divergence data for the 10m wind field. F_SCV is a direct model outputs of FESOM given as monthly averages. Monthly averaged Eddy Kinectic Energy (F_EKE) is calculated based on the monthly averaged surface velocities following Wekerle et al. (2017).

## 3 Results

Figure 1a provides an overview of the Arctic Ocean with IBCAO ocean depth and individual regions, sectors and points of interest for the subsequent description of detailed results and discussion. An overview of the spatial distribution of mean LFQ





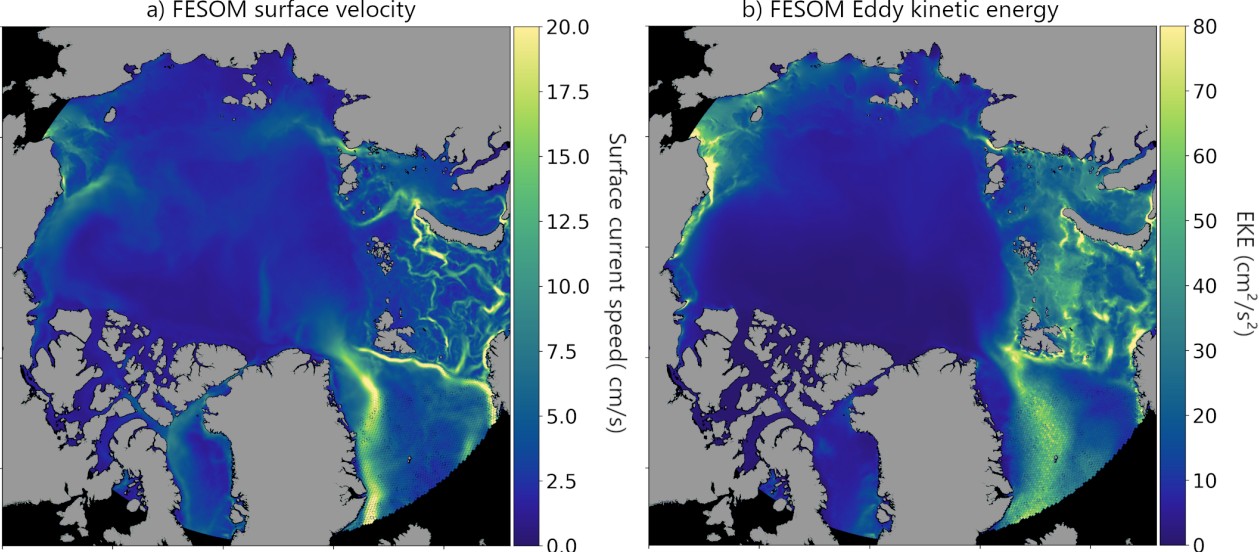

**Figure 2.** a) FESOM ocean surface current velocity (F_SCV) and b) FESOM eddy kinetic energies (F_EKE), NOV-APR, 2002/03 - 2015/2016.

based on the new lead climatology of Reiser et al. (2020) given in Figure 1b. There are several distinct spatial patterns in the observed lead climatology, which confirms the findings of Willmes and Heinemann (2016), but with far more detail. Increased

lead frequencies with values of up to 0.4 (pixel is covered by a lead in 40% of cases) are mainly found on the shelves, along the shelf break and alongside of channels and distinct bathymetric features (Fig. 1b). Several lead "hot spots" can be identified by darker shading that were hitherto unknown or at least not described in detail (see section 3.2). The most dominant spatial lead patterns are mainly found in Fram Strait (FS) and in the Barents / Kara Seas (BK), thus close to the marginal ice zone, where the ice pack becomes less dense due to the increased influence of ocean swell and waves (e.g. Bennetts et al., 2022). The lead

data shown here indicate, however, that also in the BK region we can spatially distinguish regions with pronounced LFQ that are associated with sea floor channels or ridges. Also, e.g., the northern edge of the Yermak plateau (northwest of Svalbard, Point 1 in Figure 1) shows an increased lead activity as compared to the surrounding, which gives rise to the assumption that mechanical or thermodynamical sea-ice weakening due to ocean currents influences sea-ice dynamics in this region. Similar patterns are also found around Franz-Josef Land (2), in St. Anna Trough (3) and west of Novaja Semlya (4). The Beaufort Sea

(BS) and its associated gyre are characterized by a significantly increased LFQ (>0.3) as compared to the central Arctic Ocean, where LFQ can drop down to below 0.05. The Laptev and East Siberian Seas (LS, ES) are mostly characterized by the extended fast-ice area and large flaw polynyas. Additionally, the shelf break as well as several small islands and shoals pop out in Fig. 1b with very high LFQ values (>0.4 in polynyas and over shoals). In the Chuckchi Sea (CS) we also find a distinct spatial pattern in the lead climatology that is obviously influenced by the sea floor. A first explanation of these findings is provided by

the results of FESOM simulations of ocean surface current velocities and EKE (Fig. 2a, b), showing that higher values in both



FESOM variables in many places occur with increased lead frequencies. This agreement is noteworthy especially because the two data sets are independent of one another (satellite data and simulation). The F_SCV clearly show what is usually referred to as the Arctic Boundary Current (Aksenov et al., 2011) with branches crossing the channels of the Barents Sea and entering the Siberian Sea via Vilkitsky Strait (5). Some of the spatial structures exhibited by these current branches resemble the lead

patterns. For example, along the shelf break north of Svalbard, at the eastern edge of St. Anna Trough, in Vilkitsky Strait or in the Northern Chuckchi Sea, which gives rise to the hypothesis that ocean currents are an important driver for the mean susceptibility of sea ice lead occurrence. While increased current speeds are mostly associated with high F_EKE values the latter are more tied to bathymetric features and ocean depths above -1000 m.

### 3.1 Sea-ice leads, bathymetry and ocean currents

A more detailed perspective on the relation of lead frequencies, ocean depth and ocean currents is presented in Figures 3, 4 and 5 for three different sub-regions and transects.

### 3.1.1 Barents and Kara Seas

In Figure 3a, we show mean LFQ, IBCAO ocean depth, mean F_SCV and mean F_EKE in the Barents Sea (compare red rectangle around BK in Figure 1a). The spatial overview reveals the above-mentioned patterns in more detail. In many places

high LFQ values are associated with strong bathymetric gradients, increased current speeds and high F_EKE. Ocean depth in the Barents Sea ranges between -400 m and -100 m in the presented subset and is characterized by several elevations and channels with outlets towards the shelf break north of Svalbard (Point 1) and Franz-Josef Land (2), where ocean depth drops immediately to less than -1000 m. The eastern slope of St. Anna Trough (3) shows moreover a branch of significantly increased current speeds and a band of LFQ values above 0.4, which means that the sea ice in this region is covered by leads in more

than 40% of the time in the winter period from November to April. The highlighted transect (red) allows for a more detailed comparison in Figs. 3e and 3f. Reaching from northwest to southeast through the Barents Sea, the LFQ transect exhibits local maxima right above the indicated seafloor elevations, which reach from about -350 m up to -100 m in this region (Fig. 3e). At the northwestern coast of Novaya Zemlya the LFQ maximum is found over the slope rather than on top of the shelf (4). At this position the LFQ maximum is neighboured by strong maxima in F_SCV and F_EKE by the FESOM model (Fig. 3f).

The other seafloor elevations along the presented transect (e.g., at 69 km, 207 km, 345 km) are less accompanied by F_SCV or F_EKE maxima as by LFQ.

### 3.1.2 Laptev Sea

Similar features are found in a close-up for the Laptev Sea (Fig. 4, compare red rectangle around LS in Figure 1a). Most prominent here are the flaw polynyas and the extended fast ice area, which are characterized by high and low lead activity,

respectively. But increased LFQ values are also found in the outflow region of Vilkitsky Strait (5), between the Taimyr Peninsula and Bolshevik island, and along the shelf break (6). The chosen transect crosses Vilkitsky Strait up to the shelf and continues





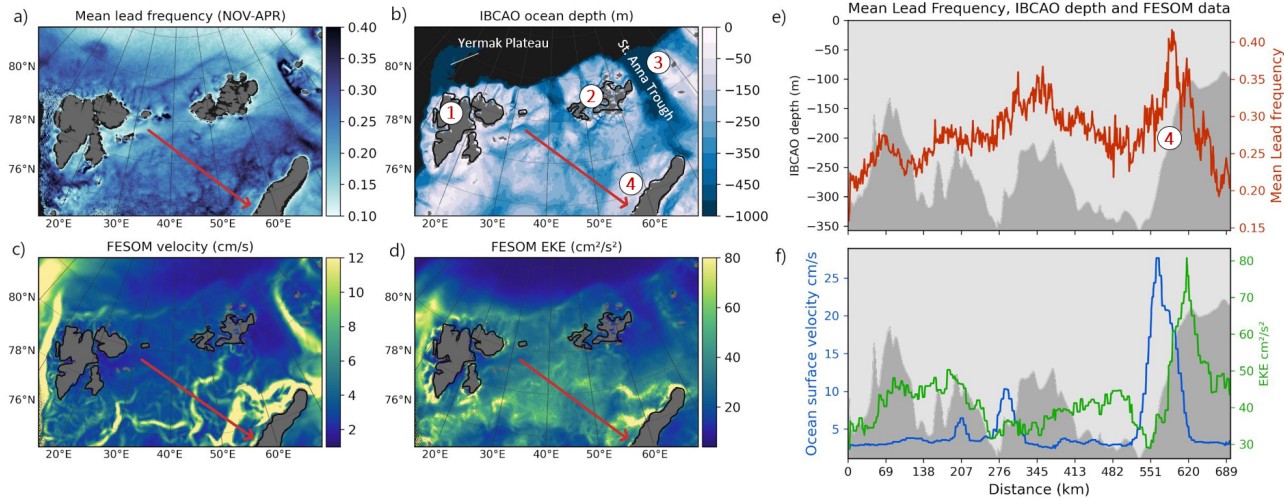

**Figure 3.** a) Mean wintertime lead frequency 2002-2016, b) IBCAO ocean depth (Jakobsson et al., 2020), c) mean FESOM surface current velocity (F_SCV) and d) mean FESOM EKE (F_EKE) in the Barents Sea; transect (red line) values: e) mean lead frequency (red) and ocean depth, f) FESOM current velocity (F_SCV, blue) and FESOM EKE (F_EKE, green), compare red box A in Figure 1a.

over the shelf break towards the deep ocean. Here, increased LFQ is found at the start of the transect southeast of Bolshevik Island where a flaw polynya is found, at the southern slope of Vilkitsky Trough and over the shelf break (Fig. 4e). Except from the flaw polynya, we find the mentioned LFQ maxima to be associated with maxima in both, F_SCV and F_EKE. Local maxima for both parameters are found on the slope of Vilkitsky canyon as well as over the shelf break, indicating that the ocean plays a crucial role in shaping the sea ice stability in this region. The importance of the Vilkitsky canyon in transporting water masses was documented in several studies (e.g.: Harms and Karcher, 2005) with the general surface circulation in the Laptev Sea being characterized by an eastward flow that causes an inflow of saline water masses from Vilkitsky Strait (Janout et al. 2020). While the inflow itself does not explain increased lead frequencies in this region and over the shelf break, intensified currents and tide-induced shear (Janout et al., 2015; Janout and Lenn, 2014) might be a driver for frequent sea-ice break-up. Thus, changing water masses and strong surface gradients in Vilkitsky Strait and over the Laptev Sea shelf break potentially represent favourable conditions for the formation of leads. The overall lead patterns in the Atlantic and Siberian sectors of the Arctic Ocean with increased lead frequencies over seafloor channels, ridges and along the shelf-break might therefore well be related to the structure and pathway of the Arctic Circumpolar Boundary Current (Pnyushkov et al., 2015, Aksenov et al, 2011).

### 3.1.3 Chuckchi Sea

In the Chuckchi Sea (Fig. 5, compare red rectangle around CS in Figure 1a) and around Wrangel Island we find some point-shaped areas of high LFQ and a linear band of slightly increased LFQ reaching from the north-eastern edge of Bering Strait towards Northwest. The IBCAO subset indicates that the latter pattern is associated with a shallow channel in this region,





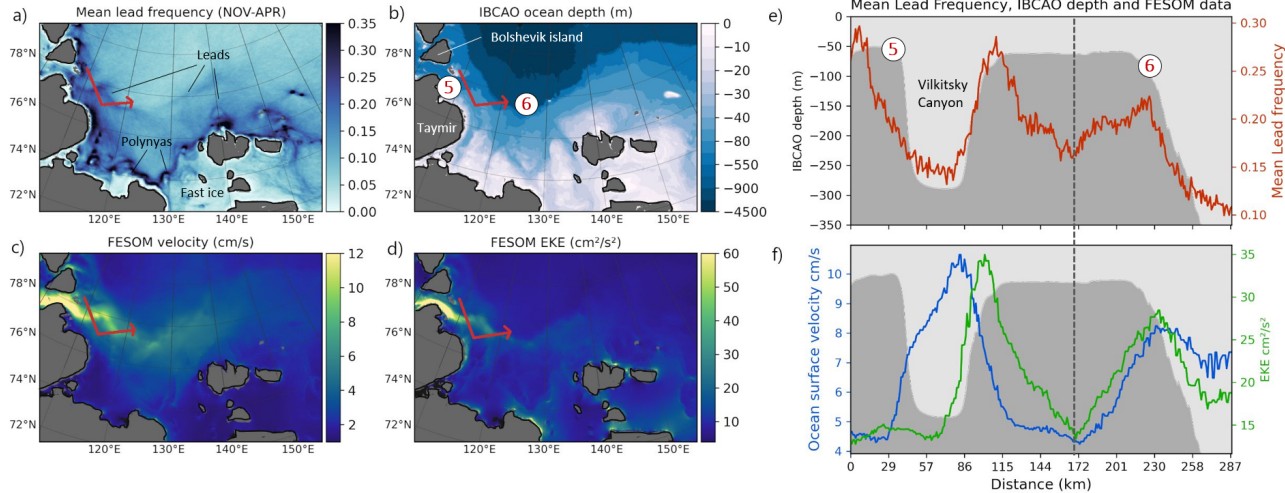

**Figure 4.** As in Figure 3 but for the Laptev Sea, compare red box B in Figure 1a.

i.e. Herald Canyon (7), and the southwestern slope of Herald Shoal (8), while the point-like LFQ maxima are found north of Point Barrow (9), over Hanna Shoal (10) and next to a small island east of Wrangel. The ocean current velocities show two branches of increased speed reaching up north alongside of Herald Shoal. F_EKE values are increased next to the coast and also in the band of increased LFQ on the western side of Herald Shoal. The profile in this subset crosses Herald Canyon, continues over Herald Shoal and ends in the central channel of the Chuckchi Sea. It indicates the strongly increased LFQ on

the southwestern slope of Herald Shoal that is part of the band of higher LFQs mentioned above and with a pronounced peak right over Herald Shoal (Fig. 5e). The latter is accompanied by increased F_EKE in the ocean data, while the LFQ peak at the slope shows maxima for both, F_SCV and F_EKE (Fig. 5f). Towards the Central Channel, LFQ as well as F_SCV and F_EKE increase coincidently. The indicated lead patterns can be attributed to the general circulation in the Chuckchi Sea, which is characterized by a broad northeasterly flow following the topography with some regions of intensified currents, e.g. in Herald

Canyon (Winsor and Chapman, 2004; Stabeno et al., 2018).

### 3.1.4 Overall influence of surface currents and EKE

In order to pinpoint the Arctic regions, where the ocean (F_SCV and F_EKE) has the largest influence on the observed lead frequencies, we calculated what we refer to as the Conicident Percentile Exceedance (CPE), which is the percentile value that is exceeded in both data sets (LFQ and F_SCV or LFQ and F_EKE, respectively) at a certain position (Figure 6). This metric

allows us to identify the regions, in which there is indication for the ocean to be as a significant driver for lead dynamics. The resulting map shows distinct patterns with CPE values over 90 in several regions (i.e. values are above the 90th percentile in both datasets). Most dominant are the Fram Strait, The Barents and Kara Seas, as well as the coast of Alaska, north of Point Barrow. Several bathymetric features can be recognized in both maps, e.g.: Herald Canyon in the Chuckchi Sea, the Vilkitsky





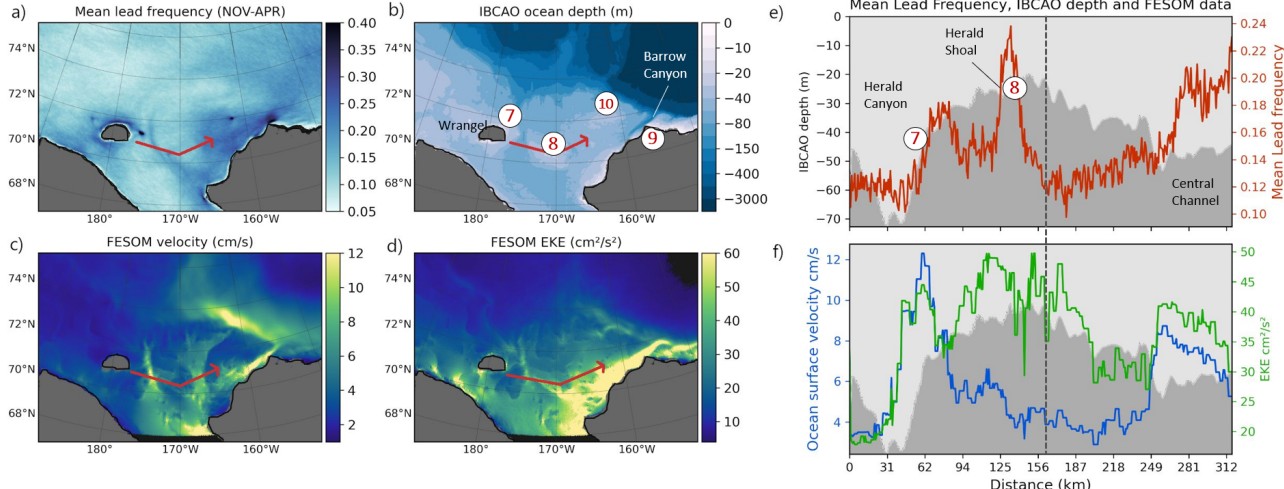

**Figure 5.** As in Figure 4 but for the Chuckchi Sea, compare red box C in Figure 1a.

Canyon in the western Laptev Sea or the slope of St. Anna Trough in the Kara Sea. In Fram Strait, we find high CPE for

LFQ/F_SCV in a broad band between Greenland and Svalbard (Figure 6a), while the influence of F_EKE on leads seems to be limited to the marginal ice zone / ice edge in this region. In the Barents Sea we find individual current branches that are associated with high LFQ values. Many more interesting details can be seen in the two maps in Figure 6 that provide insight into how F_SCV and F_EKE affect the long-term lead dynamics individually and in which regions the stability of sea ice seems to be more prone to processes in the ocean.

### 3.1.5 Spatial lead patterns and winds

The presented results indicate that the effect of the ocean on sea-ice lead dynamics is most obvious in the long-term, that is, in the climatology. This is argued for by comparing IBCAO ocean depth with the mean fields of LFQ, F_SCV and F_EKE. As far as the atmosphere is concerned, average wind speed and shear (not shown) provide no indication of driving the observed patterns in the lead climatology. However, the mean wind divergence (Figure 7) shows a dominant region of positive divergence

centred around the Beaufort Gyre. This divergence pattern with maxima especially in the southern Beaufort Sea, is well aligned with the Beaufort Sea LFQ maximum indicated in Figure 1b. This broad region of generally increased lead occurrences might therefore be influenced by the mean divergence of the wind field rather than by ocean currents. Also, Fram Strait is characterized by a positive wind divergence in the long-term mean that can consequently add up to the influence of enhanced ocean currents on average lead frequencies in this region. In general, however, winds seem to have a larger effect on the temporal dynamics

of leads rather than on the mean spatial patterns. This will be presented in detail in section 3.3. First, however, an overview of the general spatio-temporal lead variability is presented.





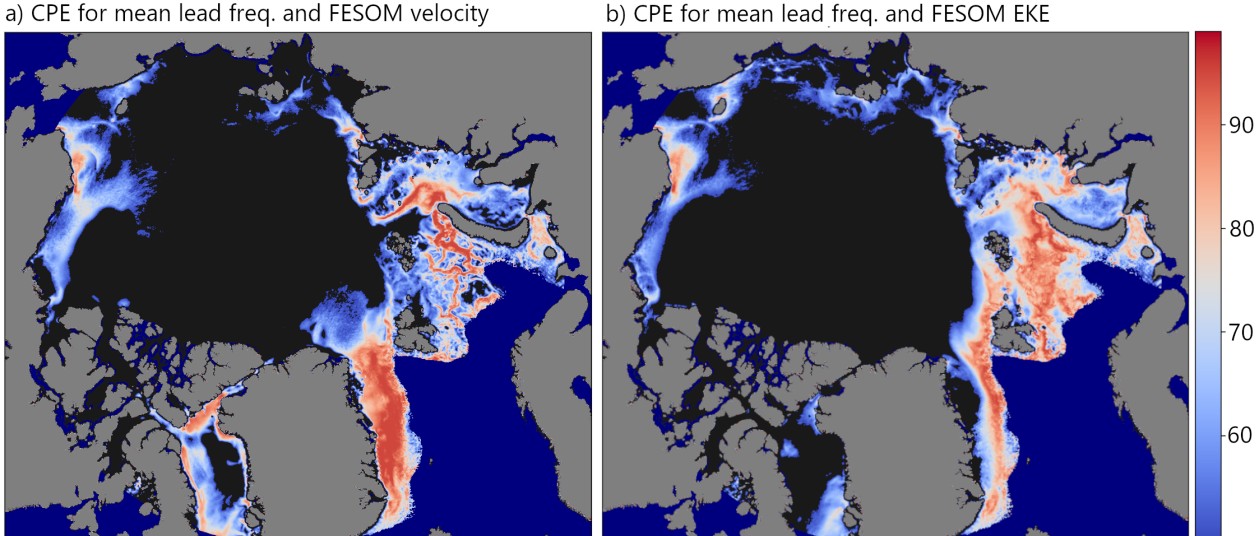

**Figure 6.** The Coincident Percentile Exceedance (CPE) for the mean wintertime lead frequency 2002-2016 and a) FESOM surface current velocity (F_SCV) and b) FESOM EKE (F_EKE). Values indicate the value percentile per pixel that exceeds in both data sets coincidently.

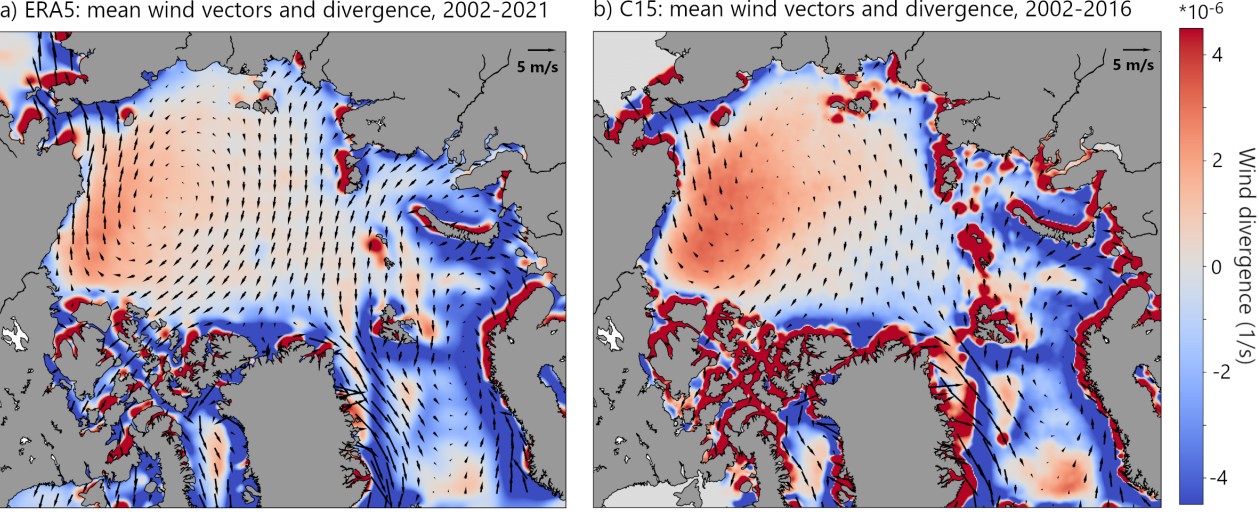

**Figure 7.** Spatial fields of mean wind divergence (in $10^{-6}s^{-1}$) and mean horizontal wind vectors for a) ERA5, 2002-2021 (Hersbach et al., 2020) and b) C15, 2002-2016, for November to April. Wind vectors are shown with a spatial distance of 15 grid points for ERA5 and 10 grid points for C15, respectively (wind vector scale in the upper right corner)





## 3.2 Spatio-temporal lead variability

The MODIS lead climatology (Reiser et al., 2020) allows for an in-depth overview of wintertime sea-ice dynamics during the last two decades. As shown in Figure 8a, no overall trend is present in pan-Arctic monthly lead fractions (LFA). However, a
significant seasonal and inter-annual variability can be noted with the tendency of a generally lower lead fraction at the end of winter (April) with frequency values of 0.1 on average compared to 0.15 in November. This is a reasonable finding considering the ice becoming thicker, more compact, and thus less mobile throughout the winter months. Subfigures 8b – 8i show the inter-annual and seasonal variability of lead fractions for individual regions (see map in Fig. 8a). None of the presented regions exhibits a significant trend. The magnitude of LFA differs between regions with the largest area covered by leads found in the
FS sector (>0.5, note the different scales for individual regions). A strong seasonal variability is present in all regions, which is also indicated by the seasonal evolution of lead fractions per region (right panel for each region). While the change in area covered by leads is less pronounced in the FS, CN and BK sectors throughout winter, all other regions are characterized by a continuous decrease in lead fractions from November to April. In the central Arctic (CA) lead fractions are the lowest (<0.1 on average) and rather stable from November to March, but a significant drop is present from March to April, when lead fraction
here is on average only around 0.05. Outliers in monthly lead fractions in all sectors indicate that strong anomalies are found only on the monthly scale, while there is no full year with extremely anomalous lead fractions on the pan-Arctic and on the regional scale.

This finding is presented in more detail in latitude-averaged monthly LFQ anomalies in 5° longitude bins for Nov 2002 to April 2021. Figure 9a shows the temporal evolution of LFQ anomalies in sea-ice regions between 70°N and 90°N across
longitudes in the Arctic Ocean (the coastlines below the Figure indicate the respective position across longitudes). Also here, trends cannot be identified. The most interesting feature here is an indication of significant lead events that are expressed through strong positive anomalies (red dots). Several of these events can be identified from the diagram across all longitudes while the Siberian sector of the Arctic (90°E - 180°E) is generally characterized by a variability that is less pronounced than in other sectors. Two events are exemplarily extracted and shown as monthly lead anomaly maps. In Figure 9b, the monthly
LFQ map shows strong anomalies north of the Canadian Archipelago with values of up to 15% above average (see marker b for comparison in Fig. 9a). An extreme lead event in the Beaufort Sea with similar anomalies is shown in Figure 9c (marker c in Fig. 9a), which resulted from a persistent high-pressure system over the central Arctic Ocean causing and intensification of the Beaufort Gyre (Babb et al., 2019). While the here presented short-term, i.e. monthly, variability of sea-ice leads is surely driven by many factors we here want to analyse the contribution of winds, i.e. wind speed and wind divergence.

## 3.3 Sea-ice leads and winds

To get a better insight into the driving mechanism for the temporal lead variability discussed above we investigate the grid-point correlation of monthly LFQ with wind from ERA5 and C15 model data, respectively. No significant correlations are found for the wind speed and wind shear (not shown). The Pearson correlation coefficient for monthly LFQ with wind divergence is shown in Figure 10 for ERA5 and C15, respectively. The latter is only available for the period 2002-2016. Only significant





**Figure 8.** a) Time series of daily (grey) and monthly (green) Arctic lead fractions, all areas: Inset map indicates individual regions. b)-f) Lead fractions for individual regions throughout years and months for regions indicated in inset map: Fram Strait (FS), Barents and Kara Seas (BK), Laptev Sea (LS), East Siberian Sea (ES), Chuckchi Sea (CS), Beaufort Sea (BS), Canadian Arctic (CN), Central Arctic (CA). Left panels: box-whisker plots for individual years with median, the 25 and 75% percentiles as the box, the 10 and 90% percentiles are plotted as error bars. Right panels: as left panels, but for individual months.


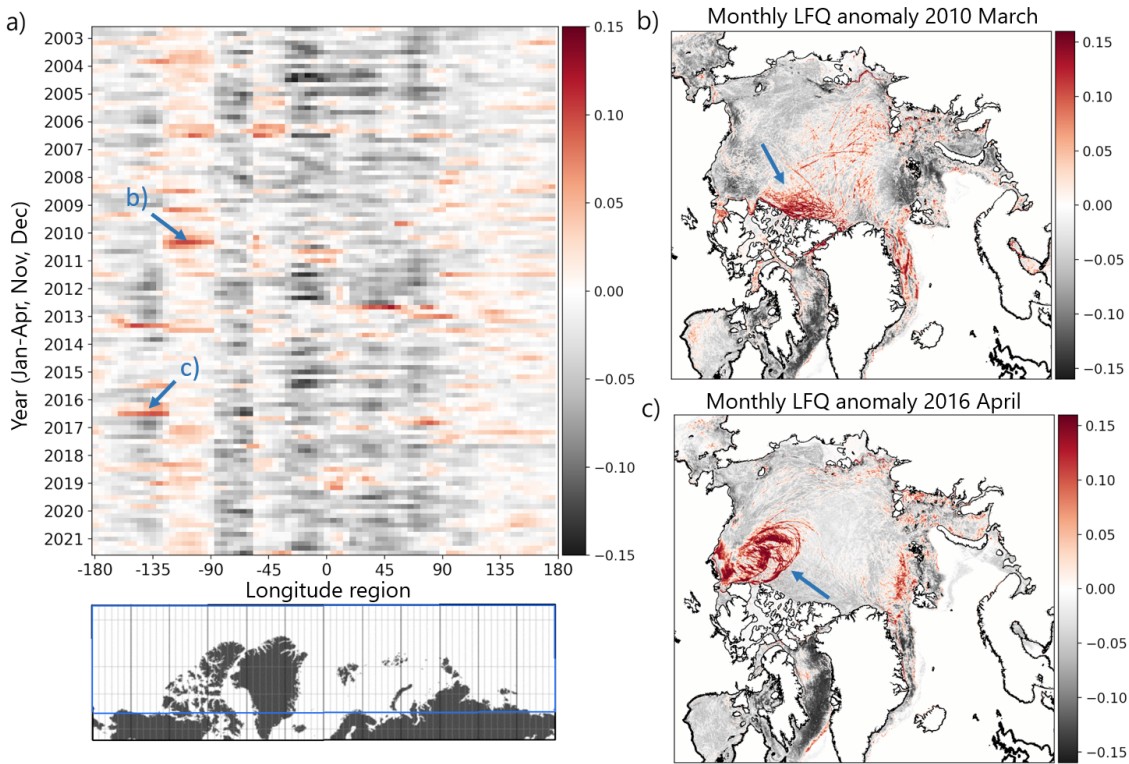

**Figure 9.** a) Latitude-averaged (70-90°N) monthly LFQ anomalies in 5° longitude bins for Nov 2002 to April 2021. Only the months Nov, Dec, Jan, Feb, Mar and Apr are shown, b) monthly lead frequency (LFQ) anomaly for 2010, Mar, with a very strong positive lead anomaly north of the Canadian Archipelago and c) monthly LFQ anomaly for 2016, April, underpinning the strong positive lead anomaly in the Beaufort Sea that is also given in the overview in a).

sectors are considered in both maps (p-values > 0.05). There are clearly two main regions, where a significant correlation between wind divergence and lead dynamics can be identified: Both, in the Beaufort Sea and in Fram Strait, the variability of monthly wind divergence correlates with the mean monthly lead frequency. While the correlation coefficient is generally only in the range between 0.5 and 0.7, the clustering of values in the two mentioned regions indicate that wind divergence is a significant driver of lead dynamics here. For the C15 data, there is a maximum correlation in the approximate center of the Beaufort Gyre (Fig. 10b), while this is less pronounced with ERA5 data (Fig. 10a). Towards Fram Strait, correlations are increasing towards South.

No significant correlation is found in the central and Siberian Arctic, suggesting that on average winds play only a minor role in the regional lead dynamics for these sectors. The Barents and Kara Seas indicate some sectors with significant correlation with a maximum at the ice edge. Two points are indicated (1: central Beaufort Sea, 2: Fram Strait) from where time series are extracted to show the temporal evolution of monthly values of LFQ and wind divergence from ERA5 and C15, respectively. Figure 11 shows the evolution of monthly averages of these data at points 1 and 2 in Figure 10 for November to April, 2002-





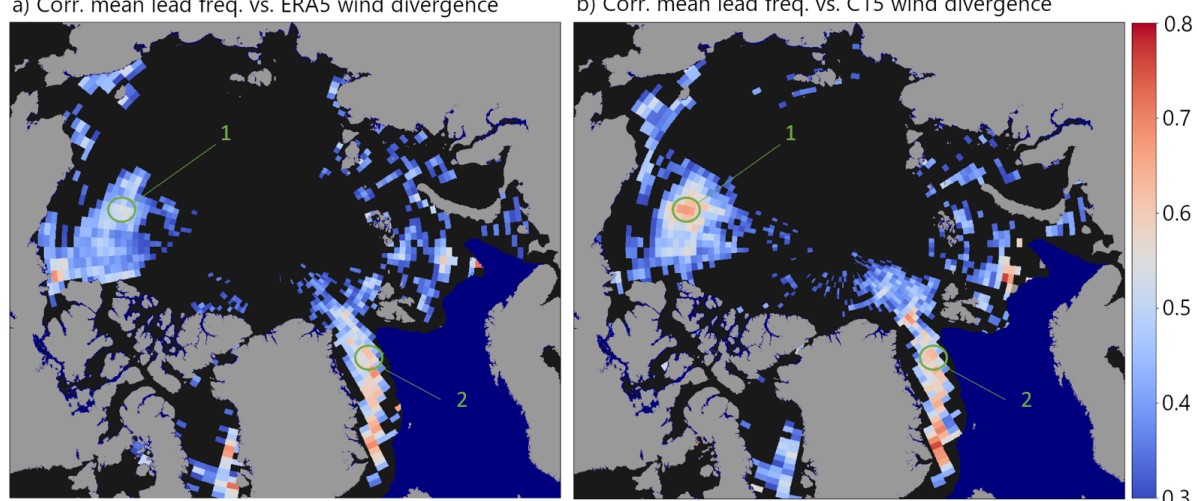

**Figure 10.** Pearson correlation for monthly values of mean lead frequency and monthly mean wind divergence from a) ERA5 (Hersbach et al., 2020), 2002-2020 (Nov-Apr) and b) C15, 2002-2016 (Nov-Apr). Coefficients are only shown for p-values<0.05.

2021 (ERA5) and 2002-2016 (C15). The comparison of time series suggests that peaks in the mean LFQ (grey) are very often accompanied by high average wind divergence (blue and orange). This is also confirmed by the associated scatterplots on the right-hand side of the time series panels, where a positive correlation between wind divergence and LFQ is indicated for both ERA5 and C15 data, respectively. In Point 1 (Beaufort Sea), the correlation coefficient is 0.67 for LFQ with C15 data and 0.54 for LFQ with ERA5 data. In Point 2 (Fram Strait) these values are 0.69 and 0.61, respectively. Figure 7 puts this in context with the fields of mean divergence and mean horizontal wind vectors from ERA5 and C15 data sets. If the coastal regions are not considered (where the effects due to friction and topography dominate), the main maximum in mean wind divergence is found in the Beaufort Sea. This maximum is associated with the mean anticyclonic circulation of the Beaufort high and is more pronounced in the C15 data. The mean wind vector fields reflect also the Transpolar Drift from the Laptev Sea to Fram Strait and the southward winds with high directional constancy over Fram Strait.

## 4 Discussion

With the perspective of a changing Arctic, knowledge about the drivers of wintertime sea-ice dynamics is key. In this context, insight about where and when leads are forming and an overview about the conditions that favour the formation of leads are essential. When extreme sea-ice break-up events are observed, the focus is most often set on explaining the driving mechanisms (Rheinlaender et al., 2022; Babb et al., 2019), which are generally strong winds and/or waves. Also in some studies, the attribution of overall anthropogenic influences on extreme events in Arctic sea ice is analyzed (Kirchmeier-Young et al., 2017). In any case, the drivers for sea-ice dynamics on seasonal and inter-annual time scales are found in ocean processes and in the

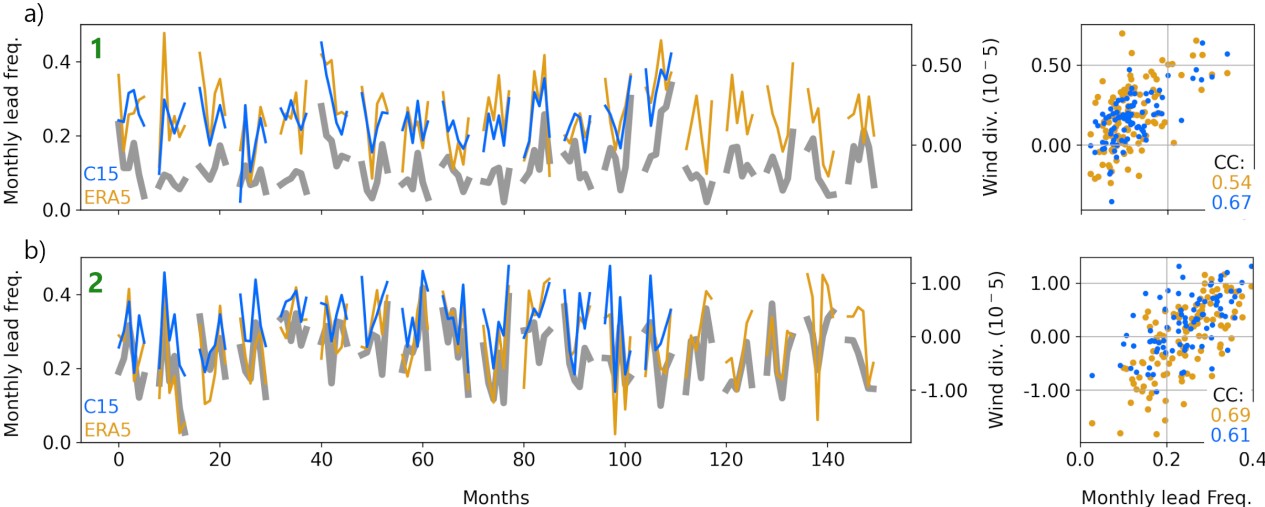

**Figure 11.** Time series of monthly and sectoral averages of lead frequency (grey) and wind divergence from ERA5 (Hersbach et al., 2020; orange) and C15 (blue) as well as associated scatterplots for a) Point 1 and b) Point 2 in Figure 10. Correlation coefficients are indicated in the scatterplots for LFQ with ERA5 (orange) and C15 (blue).

atmosphere. However, the relative importance of oceanic and atmospheric processes for changes in the Arctic sea-ice cover is

not well established (Liu et al., 2022). In this study, we therefore present a first attempt to describe and untangle the influence of both drivers spatially and temporally.The drift of sea ice and the associated stress is directly connected to the formation of leads. The main circulation patterns of Arctic sea ice, the Beaufort Gyre and the Transpolar Drift, have shown a net strengthening during last decades, which was attributed mainly to a reduced multiyear sea-ice cover (Kwok et al., 2013, Stroeve and Notz, 2018). Younger and thinner ice is expected to be more prone to break-up and lead formation (Zhang et al., 2012). However,

we could not observe trends in monthly wintertime lead fractions in the period between 2002 and 2021 (see Figure 8). A small positive lead area trend of 3700 km$^2$ per year was reported by Hoffmann et al. (2022), however with a large uncertainty and based on a different lead climatology, which shows less of the patterns highlighted in this study. Eicken et al. (2012) describe an increase in lead frequency in the Beaufort Sea in the period of 2004–2010 in comparison to 1993–2004. They explain this finding with less multi-year ice and increased divergence (Hutchings and Rigor, 2012; Babb et al., 2022). From the data we

present here, an increase of lead frequencies in the Beaufort Sea cannot be inferred, but as our analysis does not include lead observations before 2002, not conclusion can be drawn about changes in comparison to the period prior to this year.

## 4.1 Winds and leads

The role of storms and waves on wintertime sea-ice break up is well known. We could, however, not find an overall correlation between leads and wind speed in long-term data although their relation is well documented for several case studies, at least

for shorter time scales and extreme events (e.g., Graham et al., 2019, Rheinlaender et al., 2022). Instead, we find the wind



divergence to be a dominant driver for large-scale lead dynamics and sea-ice variability. We therefore assume the influence of wind speed on large-scale lead formation to be confined to short time scales and divergent conditions, at least for cases when the ice cover is still dense and compact. On the other hand, increasing wind speeds are projected for the future wintertime Arctic in conditions with a less dense sea ice cover and consequent surface warming (Mioduszewski et al., 2018). Thus, a

potentially enhanced sea-ice loss and increasing lead fractions can be amplified by the resulting intensified winds in future. In the Barents and Kara Seas with their marginal ice zones we find high lead frequencies all over. Here, the influence of winds affects both the redistribution of sea ice and the formation of leads (Pavlova et al., 2014). From the ocean side, the advection of heat through winds can trigger a thermodynamic weakening of the sea-ice cover, which can favour a break-up in regions of strong surface gradients (compare Figure 6). We find high lead frequencies also in Fram Strait, where ice export is directly

connected to higher southward ice drift velocities, due to stronger geostrophic winds (Smedsrud et al., 2011). Our findings show that also here, the enhanced lead activity can be attributed to strong ocean surface currents (in the climatology) and to wind divergence on seasonal scales (Figures 6 and 10). In the Beaufort Sea, we find a large region of enhanced lead activity around the Beaufort Gyre, that shows significant correlation with wind divergence. High sea-ice divergence in this region is reported also by Spreen et al. (2017) based on RADARSAT satellite data. The finding that shifts in sea ice cover in the Beaufort

Sea are primarily wind driven was also reported by Frey et al. (2015). Qu et al. (2021) discuss a positive interannual trend in the April lead area of the Beaufort Sea and close relation to enhanced ice motion driven by an enhanced Beaufort High and persistent easterly winds. This reported monthly trend cannot be seen in our annual time series (Figure 8g).

## 4.2   Ocean and leads

The spatial patterns in mean lead frequencies (Figure 2a) seem to be significantly affected by ocean bathymetry and the Arctic

Circumpolar Boundary Current (Aksensov et al., 2011; Pnyushkov et al., 2015), especially with its Barents Sea branch and its pathways along the shelf break. While we can assume that it's not the heat of the Atlantic water directly that triggers increased sea-ice break up, it might well be intensified currents and tide-induced shear (Janout et al., 2015; Årthun et al., 2019). Tides are not included in the FESOM simulations, which means that the contribution of this component could even enhance the suggested governing role of the ocean on preconditioning the sea-ice stability. Holloway and Proshutinsky (2007) show that

tides are capable of enhancing the loss of heat from Atlantic waters. The impact of tides on sea ice is characterized by an enhanced ocean heat flux that causes a general sea-ice thinning and thereby makes the ice more prone to lead formation. This suggests that a lot of the lead patterns that can be seen in Figure 2a are due to so-called tidal leads. The fine details in the spatial lead patterns that can be seen in our lead climatology (see Figures 3, 4, 5) have probably the potential to reveal much more of what is happening in the ocean and about the pathways and branches of water masses that could hitherto only be described

with model and mooring data. For the eastern Arctic Ocean, i.e. the Laptev Sea and northern Kara Sea, an increased oceanic heat flux from intermediate-depth warm Atlantic Water to the surface mixed layer and sea ice was reported by Polyakov et al. (2020). Such a warming at the ocean surface has the potential for a thermodynamic weakening of sea ice in these regions, which can add up to the already present dynamic weakening through changing water masses and strong surface gradients. Consequently, while trends cannot be observed in the eastern Arctic Ocean in the present study, lead fractions are likely to





increase here when increased oceanic heat fluxes persist. In the Chuckchi Sea, the ocean currents and branches described by
e.g. Stabeno et al., (2018) clearly resemble the observed spatial lead patterns. Herald canyon, Barrow canyon as well as Herald
and Hanna Shoals are the main bathymetric features in this region that govern the ocean currents. We argue here that surface
gradients and eddy kinetic energies in these regions are responsible for a frequent formation of sea-ice leads at or next to these
structures. The interaction between ocean and leads occurs in two ways. As shown here, the ocean acts as a preconditioner for

sea ice break-up, the resulting leads will in turn affect ocean stratification through new ice formation and eddy formation with
an associated horizontal redistribution of salinity anomalies (Cohanim et al., 2021). Recently, interactions between ocean-ice
heat fluxes, sea ice cover, and upper-ocean eddies were discussed and highlighted as a missing feedback in current climate
models (Manucharyan and Thompson, 2022).

### 4.3  Overall lead patterns and seasonal impacts

The presented lead climatology shows a lot of very interesting regional details that could not all be described and analysed
in this paper. Especially over the shelf of the Barents and Kara Seas as well as in the East Siberian and Chuckchi Seas, very
distinct but small-scale spatial patterns of mean lead frequencies indicate a significant preconditioning of the ice cover through
the ocean and a strong role of bathymetry. These features deserve a thorough in-depth analysis to get a more detailed insight
into how the ocean shapes the sea ice and into how projected anomalies of ocean processes in the future could affect the

sea-ice cover. It is also suggested that summer sea ice might be affected by deformation events of winter sea ice and the
resulting fracturing (Korosov et al., 2022; Hwang et al., 2017). Zhang et al., 2018 show that a realistic representation of sea
ice leads in prediction systems has the potential for predicting the summer Arctic sea ice extent. With the presented results
we argue that seasonal lead dynamics are mainly driven by winds (e.g. Liu et al., 2022) with the wind divergence yielding
the most significant correlations with temporal lead dynamics. The ocean, however, is highly responsible for preconditioning

the ice cover and making it more prone in some regions for thinning and break-up than in others with a distinct influence of
bathymetry.

## 5  Conclusions

We here present the first analysis of the drivers for the spatial and temporal patterns of Arctic sea-ice leads based on a 20-years
climatology from satellite observations with a 1 km$^2$ spatial resolution. No long-term trends can be identified for regional

or pan-Arctic average wintertime sea-ice lead fractions in the period from 2002 to 2021. In most regions, the monthly lead
fractions decrease on average slightly towards the end of winter. The results of our analysis reveal a strong influence of ocean
depth and associated currents on the frequency of occurrence of leads. This suggests that the ocean acts a preconditioner for
sea-ice break-up through dynamic processes mainly over the shelf break, along channels and slopes and over shoals. Moreover,
we find indication that regions with a high average wind field divergence, i.e. the Beaufort Sea and Fram Strait are more

prone to lead formation than other regions. These regions are also characterized by a significant temporal correlation of the
monthly wind field divergence and monthly lead frequencies, while wind speed and wind shear do not correlate with changes

in lead dynamics on long-term monthly time scales. We can conclude from the presented results that the ocean (in conjunction with significant bathymetric features and associated currents) as well as a high mean wind divergence can prime the sea-ice in making it more vulnerable to break-up. Thereby, leads are more likely to occur in these regions when synoptic conditions

are appropriate. While we assume events with high wind speed to contribute strongly to lead formation, we find the wind divergence to represent these conditions better on the monthly time scale. The presented spatial patterns of the used Arctic lead climatology exhibit a lot of details that can be used for a more thorough regional analysis of air - sea ice – ocean interactions.

*Data availability.* The daily wintertime lead data for the period November 2002 to April 2021 are currently under consideration for publication as annual files in NetCDF format on PANGAEA (*ArcLeads: Daily sea-ice lead maps for the Arctic, 2002-2021, NOV-APR*, DOI is
pending and will be included in the final version). CCLM data are available in the World Data Centre for Climate (https://www.wdc-climate.de/ui/entry?acronym=DKRZ_LTA_474_dsg0001). Version 4.0 of the International Bathymetric Chart of the Arctic Ocean (IBCAO, Jakobsson et al., 2020) was acquired from https://gebco.net and ERA5 data (Hersbach et al., 2020) were downloaded from the Copernicus Climate Data Store (https://cds.climate.copernicus.eu/).

*Author contributions.* SW analysed the data and drafted the main script. FS performed the FESOM simulations and EKE calculations. The
final version was prepared with contributions from all co-authors.

*Competing interests.* The authors declare that they have no conflict of interest.

*Acknowledgements.* The research was funded by the Deutsche Forschungsgemeinschaft (DFG) in the framework of the priority program "Antarctic Research with comparative investigations in Arctic ice areas" under grant WI 3314/3 and by the Federal Ministry of Education and Research (BMBF) under grant 03F0831C in the frame of German-Russian cooperation "WTZ RUS: Changing Arctic Transpolar System
(CATS)". We acknowledge the CLM Community and the German Meteorological Service for providing the basic COSMO-CLM model. This work used resources of the Deutsches Klimarechenzentrum (DKRZ) granted by its Scientific Steering Committee (WLA) under project ID bb0474. The FESOM simulations were performed with resources provided by the North-German Supercomputing Alliance (HLRN). We thank Lukas Schefczyk for performing the CCLM simulations. The publication was funded / supported by the Open Access Fund of Universität Trier and by the German Research Foundation (DFG).



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
