# Peer review of "Patterns of wintertime Arctic sea ice leads and their relation to winds and ocean currents"

_The Cryosphere, 2023_

## Referee Comment (RC1)

**Review of "Patterns of wintertime Arctic sea ice leads and their relation to winds and ocean currents"**

**Summary**

Willmes et al. present a new sea-ice lead climatology from 2002-2021 based on high-resolution MODIS imagery. They identify wintertime lead patterns in the Arctic and explore the role of ocean bathymetry and atmospheric forcing. To this end they use ocean data from FESOM and winds from atmospheric reanalysis data.

They find a potential, yet questionable, link between long-term lead patterns and ocean depth. Lead patterns appear to coincide with regions of sharp gradients in ocean bathymetry and associated currents which is suggested to precondition lead formation through mechanical – or thermodynamical weakening of the ice cover. The presented hypothesis is intriguing but lack a detailed mechanistic understanding supported by model or theoretical work.

Winds are suggested to play a role for the short-term variability, and large scale lead patterns, but can be questioned due to the coarse resolution of the atmospheric data. Finally, they present a very nice analysis of the spatio-temporal variability of sea-ice lead for different Arctic regions.

Overall, this is a nicely written and well-structured paper. Figures are clear and support the key findings and text well. There are many interesting things to unpack from this analysis, which I personally think deserves more attention. The current manuscript could easily be split into two separate papers; one on the link between ocean depth and lead patterns and another on the spatio-temporal trends in leads (including an analysis of the main drivers). This would allow you to go into more detail.

Below I outline some of my general concerns followed by specific in-text comments.

**General comments**

**Link between ocean depth and lead patterns**
Your analysis suggest a link between ocean depth and the dominant lead patterns through topographically steered ocean currents (if I understand you correctly?). You did a nice job showing how they could be connected from a statistical viewpoint. However, I have a few concerns

- The fact that you see a link between ocean depth and lead patterns does not infer causality. One could equally argue that the ocean current patterns are shaped by the sea ice drift/lead openings which is ultimately driven by the winds.
  You need to show how the FESOM fields translates into sea-ice deformation giving rise to the observed lead patterns. It would be more convincing if you could show the sea ice output from FESOM and show that it reproduces the same spatial patterns you find in the observations.

- Indeed, ocean bathymetry steers ocean currents around the Arctic basin. The vorticty input sustaining this circulation comes from the large scale wind field setting up a deep barotropic

circulation along f/H contours (here I'm neglecting thermohaline forcing to be clear). Making the distinction between topographically steered currents and winds as drivers for the observed patterns does thus not make a lot of sense to me as they are clearly linked.

- Finally, an in-depth description/analysis is lacking of how these topographically steered boundary currents (which sits at intermediate depths) are affecting the sea ice cover – either thermodynamically or mechanically. See Polyakov et al 2020 for a start (https://agupubs.onlinelibrary.wiley.com/doi/pdf/10.1029/2020GL089469)
  You discuss it briefly in 4.2, but it deserves more attention in the results since this is your key finding. This should further be backed up by results from the ocean model, e.g., can you show that these regions have higher shear-driven turbulent mixing, or enhanced ocean heat in the surface which could weaken the ice cover?

**Ocean data and FESOM model**
Your conclusions rely heavily on a single model and its ability to reproduce the observed features of the Arctic boundary circulation. I think FESOM is a valid choice, but a few references to studies where the model has been thoroughly validated would help strengthen your conclusions. I still wonder if you considered testing other models like the TOPAZ system (https://ocean.met.no/cmems) to see if you get similar results? I do realize that this would expand the paper considerably, which is why I recommend splitting it up in two papers.

**Spatial lead patterns and winds**
As you point out, the winds (particularly divergence) explain the observed lead patterns on the short timescale (weeks to months). I would argue that the long-term lead patterns are just representing the integrated effects of the winds over the short term. Thus, I find the statement about the ocean being the main driver for lead dynamics on long timescale questionable.

The fact that you are just picking up the large-scale patterns in the climatology (e.g. in the Beaufort Gyre) could be due to the coarse resolution of the atmospheric data. The resolution of the atmospheric data is 15 and 30 km2 which is much coarser than the lead frequencies (at 1 km2) and 4.5 km for FESOM outputs. Therefore, I guess you wouldn't expect to see the fine-scale patterns in the wind field, as you do in the FESOM output due to its higher resolution. This doesn't seem like a fair comparison and should at least be discussed.

In addition, the mean winds may not be the best metric when it comes to linking atmospheric variability and lead patterns. A more appropriate metric may be maximum or median winds. Or >90th percentile winds. See also the discussion in MacKenzie and Hutchings 2022.

**Trend in leads**
To me it's surprising that there are no trends in LFA when there are significant trends in Arctic sea ice drift and deformation (e.g. Spreen et al 2011 http://doi.wiley.com/10.1029/2011GL048970 , Rampal et al. 2009 http://doi.wiley.com/10.1029/2008JC005066) linked to decreasing ice thickness and mechanical weakening of the ice cover. You briefly touch upon this in the discussion (L274-281), although I think it deserves more attention. Would be nice if you can discuss this in more detail and compare with earlier findings (Wang et al 2016

http://doi.wiley.com/10.1002/2016GL068696, Lewis and Hutchings 2019 https://onlinelibrary.wiley.com/doi/abs/10.1029/2018JC014898). As mentioned, I think just looking at the trends could be a paper in its own right.

Also, I am curious how sensitive are these results to your definition of the winter season? I would encourage you to test for different definitions (e.g. JFM) and see if you get similar results.

**Specific comments**

Abstract: Add a description of the trends (or lack thereof). This is a key result, as highlighted in the conclusions, and should thus be in the abstract as well.

L1: Add the time period the lead data covers, i.e. 2002-2021

L5: ocean depth --> ocean bathymetry

L26: can you briefly describe how changes in sea ice extent (and thickness) are important for lead formation?

L28: I think I see what you are trying to say here, but it sounds like you are saying leads can be used to monitor global change, which I think might be a bit of an overstatement.

I would change to "... for monitoring Arctic climate change"

L48: can you briefly explain the physical reasoning behind the lead detection algorithm?

It might help the unfamiliar reader to understand why there is a temperature anomaly in the first place. Also, you should list some of the shortcomings of using MODIS for lead detection, e.g. *what are the smallest leads that can be observed?*

L95: what do you mean by deformation here? divergence+shear?

To clarify, do you calculate sea-ice deformation + shear from the wind data, sea-ice model output or remote sensing (RGPS)? The reference to Spreen et al. 2017 is a bit confusing.

L96-97: This is interesting considering that Wang et al. 2016 found a significant positive correlation between shear, divergence and lead area fraction. Can you discuss that more detail?

L105: Does it mean that the pixel is covered by a lead 40% of the time over the 2002/03 - 2015/16 period?

L108-109: The FS in particular is also an area of strong current velocities and thus high deformation rates. I am not convinced ocean swell and waves are the 1st order importance for high lead occurrence here.

L113: can you give some examples what you mean by thermodynamical/mechanical sea ice weakening? Increased ocean mixing preconditioning a thinner and weaker ice cover?

L115: Is 0.3 the mean value for the whole region? Please clarify.

L118: Perhaps it is worth mentioning how you differentiate between leads and polynyas.

L130: Can you briefly describe why these three regions were chosen specifically?

L155: You should note on what time scales the ocean matters. Is this true for short timescales too?

L183: Can you explain why you choose to use the CPE approach rather than just showing correlations (as in Fig. 10 for the winds).

L198: can you comment on how atmospheric resolution impact this statement? The resolution of the atmospheric variables are 15/30 km2 which is much coarser than the lead frequencies (at 1km2) and 4.5 km for FESOM outputs. See my general comment.

L202: How do you distinguish between mean winds and ocean currents when the circulation in the Beaufort Gyre is mainly wind driven? See also Lewis and Hutchings 2019 (10.1029/2018JC014898) for an overview on lead patterns in the BS.

L207: Can you comment on the large uncertainty in LFA (both for the winter-mean and monthly-mean). What is the main source of this uncertainty/spread?

Is it from measurement errors or the detection algorithm (cloud cover, etc)? Or is it the error/spread associated wit taking the temporal mean? It is possible that the large uncertainty masks a potential trend?

L218-219: Can you comment on why LFA are low in the CA? Because of thicker sea ice cover? I would be interested seeing a map of the lead area fraction climatology with the mean MYI concentration overlay as a contour.

L231: Please indicate in the text which year (i.e. in 2010 and 2016). Also I would emphasize the 2013 breakup event in the Beaufort Sea described in Babb et al. 2019 and Rheinlaender et al. 2022. Are there any papers discussing the 2010 event? If so, add a reference.

L232: I'm not sure that Babb et al 2019 actually postulate that the breakup in 2016 was due to an intensification of the BG (but rather due to high wind events; Fig. 5).
In fact, I would argue that BG spin-up is driven by changes in the sea-ice state (not the other way around). See also MacKenzie and Hutchings 2022

https://agupubs.onlinelibrary.wiley.com/doi/full/10.1029/2022GL101408

L235: For the discussion about wind speed versus wind divergence I suggest adding a refrence to MacKenzie and Hutchings 2022. They show that it's not only the magnitude of the winds that matter but direction relative to the coast. I would add that the duration of strong winds also matter for lead openings  (see Lee et al. 2023, but for a polynya)
https://tc.copernicus.org/articles/17/233/2023/tc-17-233-2023.pdf

L247: Again, do you expect this to be different if atm variables were comparable resolution?
With the coarser atm. resolution you're only capturing the large-scale wind patterns (as in the BS and FS), which may explain why you only find a correlation there?

L255-256: Can you indicate the p-value for the correlations?

L266: I would assume waves are mostly relevant in the MIZ and perhaps perhaps less relevant in the Arctic (due to smaller swell)? Please add a reference.

L274: Nice that you mention how changes in sea ice thickness and age can lead to more breakup. I would add a small paragraph about this in the introduction too.

L277: please add Wang et al 2016 and Lewis and Hutchings 2019 to this discussion.

L283: Add reference.

L290: the future

L291: ''The the Barents ...'' Sentence could be formulated better

L292-293: I am not quite sure what you mean here. Are you talking about advection of Atlantic Water? You don't show this, so you should add a reference.

L299: Add reference to Wang et al 2016.

what is meant by "shift in sea ice cover ..."? Please clarify

L302: *"This suggest that a lot ..."* This is very speculative and should be back up by more concrete evidence (i.e. model simulations).

L319: Could be interesting to discuss this light of Preusser et al 2016 (10.5194/TC-10-3021-2016), 2019 (10.1029/2019JC014976) about trends in polynya occurrence in the eastern Arctic. I am a bit curious why they find significant positive trends in polynya openings (linked to changes in sea ice morphology), while there are no trends in leads.

**Figures**

**Figure 6:** Can you note that the color map has been cut (black colors) and at what value? Could also highlight areas of statistical significance as in Fig 10?

**Figure 9:** First of all, really nice figure!

Two comments:
1) please specify what the anomalies are calculated relative to (i.e. the climatology?).
2) Would be nice if you could also indicate when the anomalies are statistically significant.

---

## Author Comment (AC1)

**Review of "Patterns of wintertime Arctic sea ice leads and their relation to winds and ocean currents"**

**Summary**

Willmes et al. present a new sea-ice lead climatology from 2002-2021 based on high-resolution MODIS imagery. They identify wintertime lead patterns in the Arctic and explore the role of ocean bathymetry and atmospheric forcing. To this end they use ocean data from FESOM and winds from atmospheric reanalysis data.

They find a potential, yet questionable, link between long-term lead patterns and ocean depth. Lead patterns appear to coincide with regions of sharp gradients in ocean bathymetry and associated currents which is suggested to precondition lead formation through mechanical – or thermodynamical weakening of the ice cover. The presented hypothesis is intriguing but lack a detailed mechanistic understanding supported by model or theoretical work.

Winds are suggested to play a role for the short-term variability, and large scale lead patterns, but can be questioned due to the coarse resolution of the atmospheric data. Finally, they present a very nice analysis of the spatio-temporal variability of sea-ice lead for different Arctic regions.

Overall, this is a nicely written and well-structured paper. Figures are clear and support the key findings and text well. There are many interesting things to unpack from this analysis, which I personally think deserves more attention. The current manuscript could easily be split into two separate papers; one on the link between ocean depth and lead patterns and another on the spatiotemporal trends in leads (including an analysis of the main drivers). This would allow you to go into more detail.

We appreciate the reviewers' comments and are convinced that they will help to focus and tighten the scope of our hypotheses and conclusions. We agree that it must be pointed out more clearly that we do not state an overall causality between small-scale ocean currents and lead formation. Therefore, we will emphasize that lead formation can be due to a list of reasons, while we here want to provide evidence based on observational data that small-scale currents might play a larger role in the mean spatial patterns of sea-ice break-up than anticipated and discussed in the literature so far.

We also realize that it is necessary to put more emphasis on coastal geometry as an additional factor contributing to lead formation. We acknowledge your comments, and we are certain that they help to strengthen our manuscript.

The revised version is restructured based on both reviewers' comments and suggestions. The Results chapter is now first showing the inter-annual and regional variabilities and focusing on trends, including individual months. Changes have been applied to all chapters with the overall goal to tighten the focus and specify the conclusions.

We don't think that the paper should be split in two. We agree, however, that detailed analyses of the links between ocean depth and lead patterns on the one hand as well as of drivers for the spatiotemporal variabilities on the other can be done as follow-up studies of the findings presented here.

Below I outline some of my general concerns followed by specific in-text comments

**General comments**

**Link between ocean depth and lead patterns**

Your analysis suggest a link between ocean depth and the dominant lead patterns through topographically steered ocean currents (if I understand you correctly?). You did a nice job showing how they could be connected from a statistical viewpoint. However, I have a few concerns

- The fact that you see a link between ocean depth and lead patterns does not infer causality. One could equally argue that the ocean current patterns are shaped by the sea ice drift/lead openings which is ultimately driven by the winds.
  You need to show how the FESOM fields translates into sea-ice deformation giving rise to the observed lead patterns. It would be more convincing if you could show the sea ice output from FESOM and show that it reproduces the same spatial patterns you find in the observations.

We acknowledge and understand your comment. We do not infer causality but rather provide observational evidence for this as a potential hypothesis. If winds are considered as the driver for the observed leads, an explanation for the given strong spatial coincidence of bathymetric features and lead occurrence needs to be given. We don't see why winds should cause the ice to break-up in the observed areas if not additional stress from below favours this process.

We add a new figure showing FESOM open water fraction, and the difference between FESOM sea ice bottom and sea ice top stress together with the CPE for lead frequency (observed) and FESOM stress difference:

[Figure]

**Figure R1_1:** *a) FESOM mean open water fraction, b) FESOM sea-ice stress difference bottom-top, c) Coincident Percentile Exceedance (CPE) of the FESOM stress difference and mean lead frequency. FESOM data are for the period 2002-2016.*

This figure shows that the open water fraction in FESOM (which is not an ideal measure of leads, admittedly) reflects the observed lead patterns only weakly and is mainly constrained to the shelf breaks (open water fraction is slightly higher here than in other regions). However, the difference between stresses at the bottom and the surface of the ice (Fig b) clearly highlight that bottom stress is dominant in many regions (positive difference), where the observed lead frequency is high. This is also shown in the Coincident Percentile Exceedance (CPE) of the stress difference and lead frequency, which is a measure of the mentioned coincidence. In the paper we show only subfigures b) and c).

See changes in section 4.1. There is also a new subection 3.2.6 (Winds vs. ocean forcing in the mean lead fields) that refers to the new figure (R1_1) and the role of ocean forcing vs. winds in shaping the mean lead patterns.

- Indeed, ocean bathymetry steers ocean currents around the Arctic basin. The vorticty input sustaining this circulation comes from the large scale wind field setting up a deep barotropic circulation along f/H contours (here I'm neglecting thermohaline forcing to be clear). Making

the distinction between topographically steered currents and winds as drivers for the observed patterns does thus not make a lot of sense to me as they are clearly linked.

We are aware that we are dealing with a fully coupled system, in which the ocean and the atmosphere are clearly linked. Nevertheless, we are certain that it is necessary to make such a distinction when the main local forcing for observed mean lead patterns is to be explained with the goal to untangle different forcing contributions.
We also do not intend to provide detailed mechanistic explanations for ocean - sea ice interactions here, which would go well beyond the scope of this paper, but rather provide arguments for a discussion about differences in spatial lead patterns and their potential causes. We consider the presented findings in this research as a starting point to trigger motivation to foster investigations of sea-ice ocean and sea-ice atmosphere interactions in highlighted regions and to identify the remaining open questions (see changes in Chapter 4). Detailed mechanistic explanations are not meant to be given here. We agree, however, that some of the statements and conclusions in this paper need to be refined and refocused. We have added these open questions to the discussions section in 4.3

- Finally, an in-depth description/analysis is lacking of how these topographically steered boundary currents (which sits at intermediate depths) are affecting the sea ice cover – either thermodynamically or mechanically. See Polyakov et al 2020 for a start ( https://agupubs.on - linelibrary.wiley.com/doi/pdf/10.1029/2020GL089469)
  You discuss it briefly in 4.2, but it deserves more attention in the results since this is your key finding. This should further be backed up by results from the ocean model, e.g., can you show that these regions have higher shear driven turbulent mixing, or enhanced ocean heatin the surface which could weaken the ice cover?

Thanks for this remark. To support our arguments of the ocean influence we now provide the stress difference between sea-ice bottom and sea-ice surface in the new Figure 10 (see also above). We try to pick up your comment here and strengthen the discussion accordingly. In general, however, we think that a thorough in-depth analysis of how the mentioned boundary currents affect the sea-ice cover is beyond the scope of this paper because it will regionally differ substantially and can be done in a separate follow-up study.
We have tried to strengthen the discussion section 4.2 following your arguments.

**Ocean data and FESOM model**
Your conclusions rely heavily on a single model and its ability to reproduce the observed features of the Arctic boundary circulation. I think FESOM is a valid choice, but a few references to studies where the model has been thoroughly validated would help strengthen your conclusions. I still wonder if you considered testing other models like the TOPAZ system (https://ocean.met.no/cmems) to see if you get similar results? I do realize that this would expand the paper considerably, which is why I recommend splitting it up in two papers.
No other models were tested for this study. Instead of splitting this study up in two papers we would love to see the modelling community picking up the results presented here to check for systematic connections between the location and strength of boundary currents and the observed lead patterns focussing on regions with high coincidences here. Also, as stated in this manuscript, it would be interesting to identify the contribution of tides.
We have included the reference of Wang et al. 2016, which has shown that FESOM can simulate the processes that cause the formation of leads if the spatial resolution of the model is sufficiently fine.

**Spatial lead patterns and winds**
As you point out, the winds (particularly divergence) explain the observed lead patterns on the short timescale (weeks to months). I would argue that the long-term lead patterns are just representing the integrated effects of the winds over the short term. Thus, I find the statement about the ocean being the main driver for lead dynamics on long timescale questionable.

We would like to thank you for this important remark. We do not find evidence in the long-term wind data for why the mean lead patterns are right where we see them.

An in-depth analysis of the role of winds in lead formation might still give a more detailed insight. We think that there's a lot of potential for detailed analysis of atmospheric data in context with lead dynamics that we hope our results will encourage, but which we think are outside the scope of this paper. E.g., tracks of passing lows, the effect of storms or polar lows and other atmospheric "events" can be looked into individually in context with our lead data. We agree, that we need to make more clear that the partitioning of local forcings is subject to strong variability and we will strengthen our discussion towards this point.

Please see also our responses to Review 2 in this context. See changes in section 4.1. There is also a new subsection 3.2.6 (Winds vs. ocean forcing in the mean lead fields).

The fact that you are just picking up the large-scale patterns in the climatology (e.g. in the Beaufort Gyre) could be due to the coarse resolution of the atmospheric data. The resolution of the atmospheric data is 15 and 30 km2 which is much coarser than the lead frequencies (at 1 km2) and 4.5 km for FESOM outputs. Therefore, I guess you wouldn't expect to see the fine-scale patterns in the wind field, as you do in the FESOM output due to its higher resolution. This doesn't seem like a fair comparison and should at least be discussed.

That's correct. Thank you. Now mentioned in 4.1.

In addition, the mean winds may not be the best metric when it comes to linking atmospheric variability and lead patterns. A more appropriate metric may be maximum or median winds. Or >90th percentile winds. See also the discussion in MacKenzie and Hutchings 2022.

Thanks for your comment.

We have added the mentioned paper (and others) to our extended discussion.

As the figure below indicates, for the correlation with lead frequencies on the monthly time scale, using daily atmospheric data and the associated variations does not make a substantial difference. We will discuss this topic more thoroughly in the revised version and also add literature about patterns of extreme winds in the Arctic (*Gutjahr and Heinemann, A model-based comparison of extreme winds in the Arctic and around Greenland, 2018,* https://doi.org/10.1002/joc.5729).

[Figure]

**Figure R1_2:** *Left: Mean C15 wind speed (ff, m/s) from daily data, middle: 90th percentile of daily ff, right: STD of daily ff, NOV-APR, 2002-2016.*

**Trend in leads**

To me it's surprising that there are no trends in LFA when there are significant trends in Arctic sea ice drift and deformation (e.g. Spreen et al 2011 http://doi.wiley.com/10.1029/2011GL048970 , Rampal et al. 2009 http://doi.wiley.com/10.1029/2008JC005066) linked to decreasing ice thickness and mechanical weakening of the ice cover. You briefly touch upon this in the discussion (L274281), although I think it deserves more attention. Would be nice if you can discuss this in more detail and compare with earlier findings (Wang et al 2016 http://doi.wiley.com/10.1002/2016GL068696, Lewis and Hutchings 2019

[https://onlinelibrary.wi - ley.com/doi/abs/10.1029/2018JC014898](https://onlinelibrary.wi-ley.com/doi/abs/10.1029/2018JC014898)). As mentioned, I think just looking at
the trends could be a paper in its own right.

Also, I am curious how sensitive are these results to your definition of the winter season? I would
encourage you to test for different definitions (e.g. JFM) and see if you get similar results.

Lewis and Hutchings (2019) did not report significant trends in the Beaufort Sea fracturing events during
winter ("*We investigated if there were any trends in lead activity in our time series of lead presence. We
found no significant trend in the total number of days with leads per winter season. There are also no
significant trends in the total number of fractures identified per winter season*", Lewis and Hutchings 2019
[https://onlinelibrary.wiley.com/doi/abs/10.1029/2018JC014898](https://onlinelibrary.wiley.com/doi/abs/10.1029/2018JC014898)). Also Wang et al., 2016 did report that
"*wintertime lead area fraction during the last three decades has not undergone significant trends*". They
found significant trends only in summer, which is the season that our data does not cover.

The figure below shows regional trends split down to individual months. Only months with linear trends
of p-values < 0.1 are shown. It is shown for example that reported trends in the Beaufort Sea

[Figure]

**Figure R1_3:** *Regional trends of monthly lead fractions. Only months and regions with p<0.1 are given.*

The trends, including individual months, are now more thoroughly discussed in Sections 3.1 and 4.3

**Specific comments**
Abstract: Add a description of the trends (or lack thereof). This is a key result, as highlighted in the
conclusions, and should thus be in the abstract as well.
Done.

L1: Add the time period the lead data covers, i.e. 2002-2021
Done.

L5: ocean depth --> ocean bathymetry
We think that both can be equally used here but changed to "bathymetry"

L26: can you briefly describe how changes in sea ice extent (and thickness) are important for lead formation?

We think that the relevance of leads as a trigger point for many feedbacks in air-sea ice – ocean interactions has been mentioned in the first few sentences of Chapter 1.

L28: I think I see what you are trying to say here, but it sounds like you are saying leads can be used to monitor global change, which I think might be a bit of an overstatement. I would change to "... for monitoring Arctic climate change".

That is a valuable suggestion. We changed that.

L48: can you briefly explain the physical reasoning behind the lead detection algorithm?

It might help the unfamiliar reader to understand why there is a temperature anomaly in the first place. Also, you should list some of the shortcomings of using MODIS for lead detection, e.g. *what are the smallest leads that can be observed?*

We have added: This concept is based on the fact that during winter in leads the relative warm ocean is exposed to a substantially colder atmosphere. Using this approach does not allow to distinguish between thin-ice and open water leads, but only accounts for the temperature anomaly.

L95: what do you mean by deformation here? divergence+shear?

Changed to" shear and total deformation"

To clarify, do you calculate sea-ice deformation + shear from the wind data, sea-ice model output or remote sensing (RGPS)? The reference to Spreen et al. 2017 is a bit confusing.

The reference to Spreen et al. (2017) is removed here.

All parameters mentioned here are calculated from the wind data

L96-97: This is interesting considering that Wang et al. 2016 found a significant positive correlation between shear, divergence and lead area fraction. Can you discuss that more detail?

Wang et al. have found these correlations for summertime (July to September) and individual regions only.

L105: Does it mean that the pixel is covered by a lead 40% of the time over the 2002/03 - 2015/16 period?

Yes. To point this out more clearly, we changed to: pixel is covered by a lead in 40% of all days during winter

L108-109: The FS in particular is also an area of strong current velocities and thus high deformation rates. I am not convinced ocean swell and waves are the 1st order importance for high lead occurrence here.

That's correct. We changed that to: due to strong currents in combination with the increased influence of ocean swell and waves

L113: can you give some examples what you mean by thermodynamical/mechanical sea ice weakening? Increased ocean mixing preconditioning a thinner and weaker ice cover?

We specified this to: mechanical or thermodynamical sea-ice weakening due to ocean current gradients and eddies (with the associated mixing)

L115: Is 0.3 the mean value for the whole region? Please clarify.

Thanks for the remark. We changed to: (values can exceed 0.3 in some regions)

L118: Perhaps it is worth mentioning how you differentiate between leads and polynyas.

Our retrieval method does not.

L130: Can you briefly describe why these three regions were chosen specifically?

We have added:

These regions were selected because bathymetry is highly variable here a lot of details can be found in the patterns of LFQ, F\_SCV and F\_EKE.

L155: You should note on what time scales the ocean matters. Is this true for short timescales too?
That is something we cannot answer here. We only show coincidences in the climatologies here.

L183: Can you explain why you choose to use the CPE approach rather than just showing correlations (as in Fig. 10 for the winds).
The CPE is a measure of the spatial coincidence. In Figure 10 the correlation is given for time series of winds and lead fractions. Spatial correlation can be calculated as an integrated value for different kernel sizes but will lose directional detail as compared to CPE, which can be calculated on a per-pixel level.

L198: can you comment on how atmospheric resolution impact this statement? The resolution of the atmospheric variables are 15/30 km2 which is much coarser than the lead frequencies (at 1km2) and 4.5 km for FESOM outputs. See my general comment.
We agree that some atmospheric datasets potentially provide a lot more potential for an in-depth analysis of the regional forcing for lead opening/closing dynamics, while the results presented here are meant to present the new lead climatology and provide first explanations and hypotheses for why leads are found where we see them in this new data set and also point towards the open questions, which your question here is a part of.
See also Figure R1_2 here in the response letter.

L202: How do you distinguish between mean winds and ocean currents when the circulation in the Beaufort Gyre is mainly wind driven? See also Lewis and Hutchings 2019 (10.1029/2018JC014898) for an overview on lead patterns in the BS.
It is correct, that there are no strong currents or EKE values in the BS, which makes our statement here obsolete. The new Figure 10 in the revised version points to some regions in the Beaufort Sea, where stress at the sea-ice bottom exceeds the stress at the sea-ice surface (mainly in the center of the Gyre).

L207: Can you comment on the large uncertainty in LFA (both for the winter-mean and monthlymean). What is the main source of this uncertainty/spread?
Is it from measurement errors or the detection algorithm (cloud cover, etc)? Or is it the error/spread associated wit taking the temporal mean? It is possible that the large uncertainty masks a potential trend?
The given spread per year is not due to a retrieval uncertainty but represents the distribution of daily lead fractions in each region per winter. We added this information to the figure caption to be clear.

L218-219: Can you comment on why LFA are low in the CA? Because of thicker sea ice cover? I would be interested seeing a map of the lead area fraction climatology with the mean MYI concentration overlay as a contour.
We agree, but we think that adding such a comparison would exceed the scope of the paper.

L231: Please indicate in the text which year (i.e. in 2010 and 2016).
Done.
Also I would emphasize the 2013 breakup event in the Beaufort Sea described in Babb et al. 2019 and Rheinlaender et al. 2022. Are there any papers discussing the 2010 event? If so, add a reference.
We've added The event from 2013 in the Beaufort Sea discussed in Rheinlaender et al. (2022) is also visible in Figure 3.

L232: I'm not sure that Babb et al 2019 actually postulate that the breakup in 2016 was due to an intensification of the BG (but rather due to high wind events; Fig. 5).

In fact, I would argue that BG spin-up is driven by changes in the sea-ice state (not the other way around). See also MacKenzie and Hutchings 2022 https://agupubs.onlinelibrary.wiley.com/doi/full/10.1029/2022GL101408
You're right, we changed to which resulted from a series of preceding events that preconditioned the ice in the Beaufort Sea to become weaker and thinner

L235: For the discussion about wind speed versus wind divergence I suggest adding a refrence to MacKenzie and Hutchings 2022. They show that it's not only the magnitude of the winds that matter but direction relative to the coast. I would add that the duration of strong winds also matter for lead openings  (see Lee et al. 2023, but for a polynya) https://tc.copernicus.org/articles/17/233/2023/tc-17-233-2023.pdf
The MacKenzie and Hutchings paper is added and the discussion is extended by the influence of coastal geometry. Thanks.

L247: Again, do you expect this to be different if atm variables were comparable resolution? With the coarser atm. resolution you're only capturing the large-scale wind patterns (as in the BS and FS), which may explain why you only find a correlation there?
See the comments above.

L255-256: Can you indicate the p-value for the correlations?
We could, but all p-values for the coloured points in Figure 11 (new) have p-values < 0.05 (see Figure caption). Thus they are significant. We've added: (Note that p-values for all coloured points in Figure 12 are < 0.05)

L266: I would assume waves are mostly relevant in the MIZ and perhaps perhaps less relevant in the Arctic (due to smaller swell)? Please add a reference.
Changed.

L274: Nice that you mention how changes in sea ice thickness and age can lead to more breakup. I would add a small paragraph about this in the introduction too.
We've added The younger and thinner ice that was observed during recent years is expected to be more prone to break-up and lead formation (Zhang et al., 2012).

L277: please add Wang et al 2016 and Lewis and Hutchings 2019 to this discussion.
We added Lewis and Hutchings 2019 here. Since Wang et al find trends only for summer, we will not mention them in this part of the discussion.

L290: the future
Done

L291: ''The the Barents ...'' Sentence could be formulated better
We've rephrased it to In the Barents and Kara Seas with their marginal ice zones we find generally high lead frequencies.

L292-293: I am not quite sure what you mean here. Are you talking about advection of Atlantic Water? You don't show this, so you should add a reference.
The sentence was not well formulated. We changed to
the influence of winds affects the redistribution of sea ice and thereby the formation of leads

L299: Add reference to Wang et al 2016.
Done
what is meant by "shift in sea ice cover ..."? Please clarify
Changed to "changes in the sea-ice coverage"

L302: *"This suggest that a lot …"* This is very speculative and should be back up by more concrete evidence (i.e. model simulations).

(L312?) It is just a hypothesis, that we want to add to the discussion here, yes. We have weakened our statement to be clear.

L319: Could be interesting to discuss this light of Preusser et al 2016 (10.5194/TC-10-3021-2016), 2019 (10.1029/2019JC014976) about trends in polynya occurrence in the eastern Arctic. I am a bit curious why they find significant positive trends in polynya openings (linked to changes in sea ice morphology), while there are no trends in leads.

We have made this statement more clear by adding the monthly trends here that are also added to Section 3.1. now.

**Figures**

**Figure 6:** Can you note that the color map has been cut (black colors) and at what value? Could also highlight areas of statistical significance as in Fig 10?

The map start only after the 50$^{th}$ percentile each (added to caption now). Statistical significance can be indirectly deducted here by the given value itself.

**Figure 9:** First of all, really nice figure!

Two comments:

1) please specify what the anomalies are calculated relative to (i.e. the climatology?).

Done

2) Would be nice if you could also indicate when the anomalies are statistically significant.

We could do so, but we think that it does not provide much additional information because it is anyway averages for 70°N -90°N shown here in 5° lon bins, which means that details on the finer spatial scale are masked by the averaging, while they still might be significant.

---

## Author Comment (AC2)

**Response to Referee 2 Comments (D. Watkins)**
We would like to thank Dr. Watkins for evaluating our manuscript and for providing constructive feedback. In this document we will address the comments point by point.

**We show referee comments in black text and our response in blue.**
**Changes in the manuscript are described in orange..**

**Review of "Patterns of wintertime Arctic sea ice leads and their relation to winds and ocean currents"**

**Overview**
The authors compare a new high resolution wintertime sea ice lead data set against modeled ocean currents from FESOM and wind from CCLM and ERA5. The results show a compelling linkage between small-scale ocean currents associated with gradients in bathymetry and regions with enhanced lead frequency. The influence of winds is examined in terms of monthly averages of velocity and divergence. The manuscript demonstrates that the new lead dataset has broad potential for increasing our understanding of ice processes.

**General comments**
The role of small-scale ocean currents is often neglected in discussions of ice motion and deformation. This study makes a strong argument that ocean currents, particularly those forced by bathymetry, are important for setting up regions of stronger ice deformation, resulting in regions of higher lead frequency. The results are based on a new dataset as well as new runs of a modern ocean model with high spatial resolution. My main concerns are with the treatment of wind forcing and with the attribution of causality for ocean currents. In the following I will pose a few questions for the authors that I think need to be addressed before the manuscript is ready for publication.

We appreciate the reviewers' comments and are convinced that they will help to focus and tighten the scope of our hypotheses and conclusions. We agree that some points must be made more clearly and will address these in our revised version.

The revised version is restructured based on both reviewers' comments and suggestions. The Results chapter is now first showing the inter-annual and regional variabilities and focusing on trends, including individual months. Changes have been applied to all chapters with the overall to tighten the focus and specify the conclusions.

Can the lead fraction dataset distinguish between leads, polynyas, and regions of low sea ice concentration? If not, how does this affect the interpretation of regions marked as having high lead frequency? I question whether the high lead fractions shown in Baffin Bay, the Barents Sea, and the Greenland Ice Tongue are conflating leads with the typically low sea ice concentrations in those regions.

We appreciate your remark. Due to our retrieval method we define a lead as a significant positive surface temperature anomaly (see Reiser et al., 2020, https://doi.org/10.3390/rs12121957). As such, it includes low-sea concentration areas (that can during winter be considered as network of leads, except from the outer MIZ). Regions that were outside the sea-ice area during year when the ice edge was further north are not included as leads in the climatology.

How does variability in the location of the ice edge affect the reported lead fractions? From the example image in Figure 2a, I see that the highest lead frequencies are found in the marginal ice zone and along coasts. Looking at the red arrow in Figure 3a, I am concerned that the lead frequencies reported near Nova Zemlya represent different numbers of years, as there is often open water for a large fraction of that transect.

As mentioned above, if an area is south of the ice edge during any year, meaning that there is open water as given by an AMSR-E SIC of 0%, we consider this as outside of the sea ice area and the Lead Fraction is zero.

What role does wind direction have on the lead fraction near coasts?  The manuscript states that there is no indication that wind impacts the location of the lead regions. The analysis focuses on monthly averaged values of divergence and wind speed and attempts to correlate wind values with lead fraction in each grid point. I think that the lack of connection between winds and lead fraction results from the analysis method.

We totally agree. We do not intend to say that "there is no indication that wind impacts the location of the lead regions" and will adjust our statements accordingly. Winds will definitely play a role in the opening and closing of flaw leads for example. We rather want to say that there is no obvious connection between mean patterns in wind fields and the observed mean location of leads. Looking at the temporal pan-Arctic scale we find monthly changes in the divergence of the wind field to be correlated with lead dynamics. That does not mean, however, that in individual regions and for certain periods and events, strong wind speeds and coastal geometry will definitely play a significant role in the opening and closing of leads, which is e.g., shown in the preprint of Jewell et al., 2023.
We want to make this point more clear in the revised version with the changes indicated below...
Changes are made in Section 4.1, see pdf_diff with changes highlighted.

The effects of winds on sea ice represents an integration of stresses upwind. The geometry of wind stress and coastlines is a critical component of preferential lead formation. In particular, coastal regions are strongly affected by changes in the wind direction even if the wind speed is constant. See for example Lewis and Hutchings 2019, Jewell and Hutchings 2022, and the preprint of Jewell et al. 2023 for analysis of sea ice lead formation in the Beaufort Sea. How does the choice of monthly averaged winds affect the interpretation? Using monthly averages removes effects of cyclones and most synoptic variability, which are especially important for  ice dynamics.  Model output is hourly, so this aspect of dynamics is captured partially–why not use higher temporal resolution for the reanalysis? It is possible that the high lead frequency seen along coastlines and near the boundaries of landfast ice corresponds with reversals in the direction of along-shore breezes as the ice is alternately pushed toward and away from the coast. This effect would not be seen in the monthly average wind speed. Consideration of the monthly standard deviation of wind velocity components or wind direction may be useful there.

We agree that many atmospheric datasets provide a lot more potential for an in-depth analysis of the regional forcing for lead opening/closing dynamics, while the results presented here are meant to present the new lead climatology and provide first explanations and hypotheses for why leads are found where we see them in this new data set and also point towards the open questions, which your question here is a part of.

**Fig. R2_1** shows the mean wind speed, 90[th] percentile and standard deviation of C15 wind speed for 2002-2016, based on daily instead of monthly data. For the correlation with lead frequencies on the monthly time scale, using daily atmospheric data and the associated variations does not make a substantial difference. Higher spatial resolutions,  in contrast, might well do so.

[Figure]

**Figure R2_1:** *Left: Mean C15 wind speed (ff, m/s) from daily data, middle: 90th percentile of daily ff, right: STD of daily ff, NOV-APR, 2002-2016.*

We will discuss this topic more thoroughly in the revised version and also add literature about patterns of extreme winds in the Arctic (*Gutjahr and Heinemann, A model-based comparison of extreme winds in the Arctic and around Greenland, 2018,* https://doi.org/10.1002/joc.5729). We will also add a new figure to the revised version (Figure 10), which underlines the effect of the ocean:

[Figure]

**Figure R2_2:** *a) FESOM mean open water fraction, b) FESOM sea-ice stress difference bottom - top, c) Coincident Percentile Exceedance (CPE) of the FESOM stress difference and mean lead frequency. FESOM data are for the period 2002-2016.*

See changes in section 4.1. There is also a new subection 3.1.6 (Winds vs. ocean forcing in the mean lead fields) that refers to the new figure (R2_2) and the role of ocean forcing vs. winds in shaping the mean lead patterns.

Can the authors rule out other possible causes for the co-location of high lead fraction and strong ocean currents? I agree with the authors that the cross sections showing enhanced currents implies that currents are a possible cause for enhanced lead formation. However, coastal interaction in combination with higher frequency wind variability (hours to weeks) may also result in higher lead frequency. I think the case could be strengthened by testing other candidate factors, and also by testing the hypothesis that high ocean currents lead to high lead fraction by looking at other regions where ocean currents are high. Are strong gradients in ocean currents always associated with increased lead frequency? If not, can the discrepancy be explained?

Thanks for this remark. It represents one of the open questions that we want to point out with our paper rather than trying to answer it with the given analysis. Answering this question in detail requires an in-depth analysis of regional specifications in current structures and potentially the influence of warm Atlantic water, which would definitely go beyond the scope of this paper. What we aim to conclude here is, however, that there is potentially a much stronger relationship in topographically steered ocean currents and lead dynamics than anticipated so far and that regional peculiarities and

detailed mechanistic processes should be accounted for in future studies. We also think that the main forcing for lead formation will be a mixture ocean currents, winds and coastal geometry with regionally differing individual contributions.

These two figures below show where high LFQ does not correspond with strong ocean currents and vice versa.

We have added these open questions to the discussions section in 4.3

[Figure]

**Figure R2_3:** *Where do we have low lead freq. and strong currents? Regions, where mean LFQ is smaller then its 30th percentile and FESOM_vel (left) and FESOM_eke (right) exceed the percentile given by color.*

[Figure]

**Figure R2_4:** *Where do we have weak currents and high lead freq.? Regions, where mean FESOM_vel (left) and FESOM_eke (right) are lower than their 30th percentiles and mean LFQ exceeds the percentile given by color.*

**Minor comments**

49 Artefact → Artifact
Done.

95 Shear is a form of deformation, what specifically is meant here? Spreen et al. 2017 discuss total deformation is the divergence and shear added quadratically, which is a standard measurement. Perhaps the authors meant to write "Total deformation".
Changed to "total deformation"

Figure 6 – CPE is a useful tool for seeing where both lead frequency and ocean current metrics are anomalous. It would be useful to also see the regions where one quantity is anomalous and the other

is not, for example where there is enhanced lead frequency with no corresponding increase in current speeds or EKE. Are there places where a strong boundary current does not appear to affect the LFQ?

See figures R2_2 and R2_3, here in the response letter. These figures are not included in the manuscript but the discussion is extended picking up on your comment and the conclusion that can be drawn from these figures.

Figure 7 – Is it known why coast effects are so strong in C15, and why the coastal winds are so different between ERA5 and C15?

Differences in the wind between ERA5 and C15 result from a different horizontal resolution and the sea ice parameterization (see Heinemann et al. 2022, DOI: 10.1525/elementa.2022.00033). In coastal areas near Greenland and in areas with complex topography these effects can be large (see Heinemann and Zentek 2021, doi: 10.3390/atmos12121635; Kohnemann and Heinemann 2021, doi: 10.33265/polar.v40.3622).
We've added this note to the figure caption

Figure 11 – date index should be in dates for the time series, not in month numbers since some arbitrary start date, as done in Figure 8.
Changed.

245 I suggest replacing "increasing towards South" with "increase southward"
Done.

**Original Review Citation**: https://doi.org/10.5194/tc-2023-22-RC2

---

## Referee Report (RR1)

Thank you for addressing the comments raised in my first review of the manuscript. I agree with the authors on the new structure starting more generally with the lead variability and trends after moving to the link between lead patterns and ocean bathymetry. It has a nice flow and feels more coherent now. It is also nice to see that you have tried to highlight the potential shortcomings in the abstract. Overall, with these new added changes, I think the manuscript is suitable for publication. I only have a few minor comments before the manuscript is finally accepted.

**General comments**

I agree that the data shows a compelling argument for the influence of ocean bathymetry on observed lead patterns. I also acknowledge that providing a detailed mechanistic understanding of the sea ice – ocean bathymetry interaction is perhaps too much to cover in this paper. Personally, I think it should be clearer that you're presenting this as a hypothesis (in the abstract) and that we don't understand the driving mechanisms in detail yet. I think this could help enhance the impact of the paper by inspiring future research (as you pointed out as a motivation in your response).

When it comes to potential mechanisms, I recently came across this paper (https://onlinelibrary.wiley.com/doi/full/10.1029/2022JC019469), which shows how ocean currents and subsurface eddies can potentially imprint the sea ice cover mechanically. I think this would be a nice addition to the discussion (Section 4.2).

In your discussion about the trends, you mention the paper by Hoffmann et al (2022) showing a small increasing trend in Arctic leads. You mention that the discrepancy could be due to a large uncertainty and different lead climatology. That's all good. However, I would still like to see a very short discussion (1-2 lines) on some of the limitations of the ArcLeads product. Is it possible that uncertainties in LAF (e.g. from clouds) could be masking potential trends?

**Specific comments**

L5: *"... influence of ocean bathymetry and associated currents on the mechanic weakening of sea ice"*

I am not entirely convinced you actually show this. In order to strengthen that argument you would need to analyze the sea ice stresses from FESOM, and show a link between ice deformation rates and ocean currents. I would rather use "preconditioning"  because we are not really sure how it works, right? Could be both mechanical or thermodynamic processes in play.

L76: have --> has

L79: The Ocean data section I would still like to have 1-2 sentences explaining why FESOM is a good choice when it comes to representing ocean currents in the Arctic. You need to convince the reader that the model is appropriate for your specific study.

L102: I am still confused about the deformation derived from the C15 wind data. "Total deformation" makes me think of sea ice, not winds. As I understand it, you're calculating the total deformation (div+shear) of the wind (not the sea ice?). However, I don't think you need to mention the total deformation since it's just the sum of shear and div?

L135: You should specify that the trend is for the entire winter season.

L149: *"is generally characterized by a variability that is less pronounced than in other sectors."*

A bit unclear what you mean by less pronounced variability. What about: "Generally shows less (interannual/monthly?) variability compared to the other sectors"

L156-159: This would fit better in the discussion. I would also consider moving the paragraph on monthly trends (from L160) to after you talk about absence of interannual trends on seasonal timescales (L135)

L167: Explain briefly why you compare lead patterns to model data, e.g. "to understand the role of the ocean..."

L187: *"which means that the sea ice in this region is covered by leads in more than 40% of the time in the winter period from November to April."*

I would use this phrasing already in the first section (L112) because it helps the reader understand what the LFQ means.

L243: Your comment to R2 that using daily atmospheric data doesn't make a huge difference compared to monthly data should be mentioned explicitly in the text (Section 3.2.5).

L244-245: The first two lines belong in the previous section. Here it sounds like you are about to talk about the ocean impact on leads, instead of summarizing the results of the precious section.

I suggest starting this section by: "To understand the role of winds in driving the observed patterns in lead formation we look at ..."

L313: *"However, we could not observe trends in wintertime lead fractions in the period between 2002 and 2021 (see Figure 2)"*

Please clarify that you're talking about pan-Arctic here since you find trends on regional scales.

Here I would also like to see a sentence on the uncertainties in ArcLeads product and its effect on the reported trends (or absence thereof), e.g. due to clouds. See my general comment.

L353: Add a reference to

L369: What do you mean by *"dynamic weakening through changing water masses"?* Please clarify

*"... while positive trends in ..."* Please clarify trends in what (sea ice?, ocean heat?)

L380: This section reads more as a conclusion. In particular, the last part highlighting open questions. If I may, I would suggest moving the "open questions" to the Conclusion as "suggestions for future work" (e.g. for the modelling community) in order to foster and inspire more in-depth research.

---

## Referee Report (RR2)

**Comments on tc-2023-22**

The authors have improved the manuscript based on the reviewer comments. It is my opinion that the manuscript should be published, subject to (very) minor revisions, as detailed below.

157 - slight rephrasing would help, I had to read the sentence a couple times to follow. What quantity did Wang et al. (2016) measure? I'd replace "did report no" with "reported no".

171 - I believe you mean "Laptev Sea"? Vilkitsky Strait connects the Kara and Laptev Seas.

178 - The idea of the sentence works, but it needs to be rephrased to avoid being a run-on sentence. Perhaps "These are regions with complex bathymetry, and hence…" I think that the phrase "a lot of details" is too colloquial.

199 - Island should be capitalized

203 - no comma needed before $F\_SCV$

234 - 90th percentile is correct, as you had it originally, rather than 90% percentile

277 - no comma needed between regions and "where a significant"

278 - I'd rephrase as "In both the Beaufort Sea and in Fram Strait"

360 - Tides impact sea ice both through inducing a vertical gradient in the surface current and by enhancing the upward heat flux. You mention tide induced shear a few lines above, so perhaps all you need to do is change "tides" to "tide-driven mixing".

414 - I think the final sentence can be improved. As in the sentence in line 178, I think the phrase "a lot of details" is too colloquial. I think that reworking the last two sentences can help emphasize that your paper has demonstrated that there is useful local information provided in the lead fraction dataset, and that future work can leverage this dataset for more detailed investigations of the local forcing.

---

## Author Response (AR2)

**Response to R1**

Thank you for addressing the comments raised in my first review of the manuscript. I agree with the authors on the new structure starting more generally with the lead variability and trends after moving to the link between lead patterns and ocean bathymetry. It has a nice flow and feels more coherent now. It is also nice to see that you have tried to highlight the potential shortcomings in the abstract. Overall, with these new added changes, I think the manuscript is suitable for publication. I only have a few minor comments before the manuscript is finally accepted.

**General comments**

I agree that the data shows a compelling argument for the influence of ocean bathymetry on observed lead patterns. I also acknowledge that providing a detailed mechanistic understanding of the sea ice – ocean bathymetry interaction is perhaps too much to cover in this paper. Personally, I think it should be clearer that you're presenting this as a hypothesis (in the abstract) and that we don't understand the driving mechanisms in detail yet. I think this could help enhance the impact of the paper by inspiring future research (as you pointed out as a motivation in your response).

Good remark. We suggest the following change to the abstract…

*The presented investigation provides evidence for an influence of ocean bathymetry and associated currents on the mechanic weakening of sea ice and the accompanied occurrence of sea-ice leads with their characteristic spatial patterns. **While the driving mechanisms for this observation are not yet understood in detail, the presented results can contribute to opening new hypotheses on ocean-sea ice interactions.** The individual …*

When it comes to potential mechanisms, I recently came across this paper (https://onlinelibrary.wiley.com/doi/full/10.1029/2022JC019469), which shows how ocean currents and subsurface eddies can potentially imprint the sea ice cover mechanically. I think this would be a nice addition to the discussion (Section 4.2).

Thanks for pointing us to this reference! We've added it to the discussion (end of Section 4.2).

In your discussion about the trends, you mention the paper by Hoffmann et al (2022) showing a small increasing trend in Arctic leads. You mention that the discrepancy could be due to a large uncertainty and different lead climatology. That's all good. However, I would still like to see a very short discussion (1-2 lines) on some of the limitations of the ArcLeads product. Is it possible that uncertainties in LAF (e.g. from clouds) could be masking potential trends?

First, we want to make a correction to the regional monthly trends that we described in the revised version (Section 3.1). If a regional cloud-correction is applied, based on the relation between monthly cloud and lead frequencies, the mentioned trends with p-values <0.05 become even unsignificant (p-values increase when clouds are taken into account). Consequently, **a masking of trends through clouds can be ruled out**. We therefore need to change, however, the last sentence of the abstract and the last paragraph of section 3.1. accordingly. As other mentioned trends are not significant at the 5% level (some only at 10%) we think it's better not to point those out at all.

There is also small change in the Discussion section regarding the cloud correction (see below).

**Specific comments**

L5: *"... influence of ocean bathymetry and associated currents on the mechanic weakening of sea ice"*
I am not entirely convinced you actually show this. In order to strengthen that argument you would need to analyze the sea ice stresses from FESOM, and show a link between ice deformation rates and ocean currents. I would rather use "preconditioning" because we are not really sure how it works, right? Could be both mechanical or thermodynamic processes in play.

You are right that we don't know how it exactly works, but we do not think that replacing "influence" by "preconditioning" does make a difference here. We rather suggest changing "the influence" to "**an** influence" in order not to claim full insight into the mechanisms here.

L76: have --> has Done

L79: The Ocean data section I would still like to have 1-2 sentences explaining why FESOM is a good choice when it comes to representing ocean currents in the Arctic. You need to convince the reader that the model is appropriate for your specific study.

We added: **FESOM has previously been used also in other studies for the simulation of ocean currents in the Arctic (e.g., Wang et al., 2016; Wekerle et al., 2017)**

L102: I am still confused about the deformation derived from the C15 wind data. "Total deformation" makes me think of sea ice, not winds. As I understand it, you're calculating the total deformation (div+shear) of the wind (not the sea ice?). However, I don't think you need to mention the total deformation since it's just the sum of shear and div?

That's right, mentioning total deformation here is redundant. We changed to "shear"

L135: You should specify that the trend is for the entire winter season.

This is added now.

L149: *"is generally characterized by a variability that is less pronounced than in other sectors."*

A bit unclear what you mean by less pronounced variability. What about: "Generally shows less (interannual/monthly?) variability compared to the other sectors"
Phrasing is adjusted accordingly.

L156-159: This would fit better in the discussion. I would also consider moving the paragraph on monthly trends (from L160) to after you talk about absence of interannual trends on seasonal timescales (L135).
We do not understand this suggestion, unfortunately. The text on monthly trends is already at the end of the paragraph and the discussion picks up this information later on.

L167: Explain briefly why you compare lead patterns to model data, e.g. "to understand the role of the ocean..."
We added … "**to evaluate a potential role of the ocean in shaping the presented observations**"

L187: *"which means that the sea ice in this region is covered by leads in more than 40% of the time in the winter period from November to April."*
I would use this phrasing already in the first section (L112) because it helps the reader understand what the LFQ means.
We rephrased the specified sentence in L112 to be more specific.

L243: Your comment to R2 that using daily atmospheric data doesn't make a huge difference compared to monthly data should be mentioned explicitly in the text (Section 3.2.5).
We added: "**This also holds for when daily, instead of monthly wind data are considered**".

L244-245: The first two lines belong in the previous section. Here it sounds like you are about to talk about the ocean impact on leads, instead of summarizing the results of the precious section.
I suggest starting this section by: "To understand the role of winds in driving the observed patterns in lead formation we look at ..."
We think this is a valuable suggestion. We moved the first sentence to the end of the last section and start this section with **"To understand the role of winds in driving the observed patterns in lead formation we look at wind components and derived mean quantities.**"

L313: *"However, we could not observe trends in wintertime lead fractions in the period between 2002 and 2021 (see Figure 2)"*
Please clarify that you're talking about pan-Arctic here since you find trends on regional scales.
Here I would also like to see a sentence on the uncertainties in ArcLeads product and its effect on the reported trends (or absence thereof), e.g. due to clouds. See my general comment.
We added *pan-Arctic* here. We are referring to uncertainties and your general comment by also adding:
"***The trends in detected leads can be biased by overlying trends in cloud frequencies. This effect is, however, compensated for in our trend calculation.***"

L353: Add a reference to
Done.

L369: What do you mean by *"dynamic weakening through changing water masses"?* Please clarify
*"... while positive trends in ..."* Please clarify trends in what (sea ice?, ocean heat?)
We changed "*the already present*" (misleading, in fact) to "*a potential*".
We changed the sentence to "***Consequently, lead fractions are likely to increase here when increased oceanic heat fluxes persist.***"

L380: This section reads more as a conclusion. In particular, the last part highlighting open questions. If I may, I would suggest moving the "open questions" to the Conclusion as "suggestions for future work" (e.g. for the modelling community) in order to foster and inspire more in-depth research.
Good suggestion. We moved the open questions to the Conclusions.

**Response to R2**

The authors have improved the manuscript based on the reviewer comments. It is my opinion that the manuscript should be published, subject to (very) minor revisions, as detailed below.

157 - slight rephrasing would help, I had to read the sentence a couple times to follow. What quantity did Wang et al. (2016) measure? I'd replace "did report no" with "reported no".+
Done.

171 - I believe you mean "Laptev Sea"? Vilkitsky Strait connects the Kara and Laptev Seas.
No actually, we mean Barents Sea.

178 - The idea of the sentence works, but it needs to be rephrased to avoid being a run-on sentence. Perhaps "These are regions with complex bathymetry, and hence…" I think that the phrase "a lot of details" is too colloquial.
We guess that the problem was due a missing "and", which is now added.

199 - Island should be capitalized
Done.

203 - no comma needed before F_SCV
Removed.

234 - 90th percentile is correct, as you had it originally, rather than 90% percentile
Changed.

277 - no comma needed between regions and "where a significant"
Removed.

278 - I'd rephrase as "In both the Beaufort Sea and in Fram Strait"
Done.

360 - Tides impact sea ice both through inducing a vertical gradient in the surface current and by enhancing the upward heat flux. You mention tide induced shear a few lines above, so perhaps all you need to do is change "tides" to "tide-driven mixing".
Good suggestion. Changed.

414 - I think the final sentence can be improved. As in the sentence in line 178, I think the phrase "a lot of details" is too colloquial. I think that reworking the last two sentences can help emphasize that your paper has demonstrated that there is useful local information provided in the lead fraction dataset, and that future work can leverage this dataset for more detailed investigations of the local forcing
Done.